# A multi-adenylate cyclase regulator at the flagellar tip controls African trypanosome transmission

Sabine Bachmaier ®[1] ✉, Giacomo Giacomelli[2,9], Estefanía Calvo-Alvarez ®[3,9], Larissa Rezende Vieira ®[4], Jan Van Den Abbeele ®[5], Aris Aristodemou ®[1], Esben Lorentzen[6], Matt K. Gould ®[1], Ana Brennand[1], Jean-William Dupuy ®[7], Ignasi Forné[8], Axel Imhof ®[8], Marc Bramkamp ®[2], Didier Salmon[4], Brice Rotureau[3] & Michael Boshart ®[1] ✉

Signaling from ciliary microdomains controls developmental processes in metazoans. Trypanosome transmission requires development and migration in the tsetse vector alimentary tract. Flagellar cAMP signaling has been linked to parasite social motility (SoMo) in vitro, yet uncovering control of directed migration in fly organs is challenging. Here we show that the composition of an adenylate cyclase (AC) complex in the flagellar tip microdomain is essential for tsetse salivary gland (SG) colonization and SoMo. Cyclic AMP response protein 3 (CARP3) binds and regulates multiple AC isoforms. CARP3 tip localization depends on the cytoskeletal protein FLAM8. Re-localization of CARP3 away from the tip microdomain is sufficient to abolish SoMo and fly SG colonization. Since intrinsic development is normal in *carp3* and *flam8* knock-out parasites, AC complex-mediated tip signaling specifically controls parasite migration and thereby transmission. Participation of several developmentally regulated receptor-type AC isoforms may indicate the complexity of the in vivo signals perceived.

All cells perceive their environment and signal outside information to control adaptive and developmental processes. For parasites, perception of the host environment by physical and chemical cues is essential for survival and transmission. Digenetic parasites face challenging rapid changes of biotic environments between their reservoir and vector hosts[1]. Pathways processing extracellular signals are not conserved in phylogenetically distant kinetoplastids, including *Trypanosoma*[2,3], and the connection of identified signaling components remains to be solved. Trypanosomes shuttle between mammalian and insect hosts and are well-studied due to medical and economic impact as causative agents of neglected tropical diseases in humans and animals[4]. A phenotype termed social motility (SoMo)[5], a process similar to bacterial swarming[6], indicates intercellular communication or peer signaling in populations of unicellular trypanosomes. On semisolid surfaces, cultured procyclic insect forms of *T. brucei* migrate away from an initial inoculation point by radial

[1]Faculty of Biology, Genetics, Ludwig-Maximilians-University Munich (LMU), 82152 Martinsried, Germany. [2]Faculty of Biology, Microbiology, Ludwig-Maximilians-University Munich (LMU), 82152 Martinsried and Institute for General Microbiology, Kiel University, 24118 Kiel, Germany. [3]Institut Pasteur, Université de Paris, INSERM U1201, Trypanosome Cell Biology Unit, Trypanosome Transmission Group, 75015 Paris, France. [4]Institute of Medical Biochemistry Leopoldo de Meis, Centro de Ciências da Saúde, Federal University of Rio de Janeiro, Av. Carlos Chagas Filho 373, Rio de Janeiro 21941-902, Brazil. [5]Trypanosoma Unit, Department of Biomedical Sciences, Institute of Tropical Medicine Antwerp, 2000 Antwerp, Belgium. [6]Department of Molecular Biology and Genetics, Aarhus University, Gustav Wieds Vej 10c, 8000 Aarhus C, Denmark. [7]Univ. Bordeaux, Plateforme Protéome, 33000 Bordeaux, France. [8]Biomedical Center, Ludwig-Maximilians-University Munich, 82152 Martinsried, Germany. [9]These authors contributed equally: Giacomo Giacomelli, Estefanía Calvo-Alvarez. ✉e-mail: sabine.bachmaier@lrz.uni-muenchen.de; boshart@lmu.de

swarming[5]. Cyclic adenosine monophosphate (cAMP) is key for control of SoMo[7-9]. Individual flagellar tip localized members of a large multigene family of transmembrane receptor-type adenylyl cyclases (ACs) are involved in the process[7]. Swarm-like collective motion of trypanosomes in different tsetse tissues and organs has been observed[10,11], yet its relation to SoMo in vitro is unclear. A common mechanism for SoMo and directed migration in the tsetse midgut has been suggested based on correlated phenotypes upon deletion of phosphodiesterase *PDEB1*[9].

Similar to the flagellum of mammalian sperm[12], the trypanosome's single flagellum represents a cAMP signaling compartment, as it displays a strong enrichment of ACs and phosphodiesterases (PDEs), ubiquitous enzymes producing or degrading cAMP, respectively. The trypanosome flagellar tip is a site with particular importance for interaction with host tissue surfaces[13] and forms a distinct microdomain. This is defined by exclusive localization of a subset of flagellar proteins[14-17], including some members of the AC multigene family[18]. The cAMP microdomain concept was first described by Buxton and Brunton[19] and is now well established in different systems[20] including primary cilia[21,22]. The high spatiotemporal specificity of cAMP signaling within these micro- or nanodomains depends on signaling cascades with sensor proteins, transducers and effectors often arranged as multiprotein complexes, or signalosomes, that can include anchoring proteins such as AKAPs (A kinase anchoring proteins). Localized PDEs limit the diffusion of the second messenger and can generate cAMP concentration gradients across a cell, while PDE inhibition has been shown to result in loss of cAMP compartmentalization[23]. Within the very small cAMP nanodomains (~50–100 nm), extremely high local cAMP concentrations are possible that can specifically regulate even low affinity effectors located within the nanodomain[24].

Virtually nothing is known about cyclase regulation or downstream cAMP signaling in kinetoplastids. These organisms lack known cyclase activators, such as G protein-coupled receptors (GPCRs) and known cAMP effectors including cAMP-dependent PKA[25] or cyclic nucleotide-gated ion channels[26]. The large family of receptor-type ACs with an extracellular N-terminal domain was suggested to provide the diversity for reception of multiple signals, although no ligands of these ACs have been identified to date[27,28]. In bloodstream form (BSF) parasites that infect the mammalian host, disturbance of the intracellular cAMP concentration is critical for growth and cytokinesis[29-31]. Exploiting this phenotype by exposing BSF trypanosomes to lethal concentrations of PDE inhibitors, a genome-wide RNAi screen for cAMP resistance identified cAMP response proteins (CARPs) as candidates for novel cAMP effectors or pathway modulators in *T. brucei*[32]. Among these, CARP3 is a trypanosome-specific protein.

AC activity plays an important role in innate immunity subversion of bloodstream stage trypanosomes early in infection[33], The most abundant AC involved is encoded by subtelomeric polycistronic transcription units driving bloodstream form-specific variant surface glycoprotein expression (VSG expression sites)[34], hence named expression site-associated gene 4 (ESAG4). Trypanosomes shuttle between the bloodstream and tissues of a mammalian host and the alimentary tract and salivary glands of a tsetse fly undergoing a series of developmental transitions that result in defined adapted stages[35]. Cyclic AMP signaling was suggested to play a role in stage development due to differential expression of transcripts encoding AC isoforms in trypanosomes colonizing midgut, proventriculus or salivary glands, respectively[36,37]. During their complex journey through the insect vector, trypanosomes are in intimate contact with host tissue surfaces and have to cope with several bottlenecks[38].

Here, we provide direct evidence that a cAMP signalosome is essential for trypanosome migration in the tsetse vector. Perturbation of the composition of an AC complex at the flagellar tip is alone sufficient to abolish SoMo and tsetse salivary gland colonization. The mutant parasites retain full intrinsic developmental competence. The highly specific phenotype elicited by the trypanosome-specific AC regulator CARP3 will stimulate further dissection of novel mechanisms of cAMP signaling.

## Results

### CARP3 is a flagellar tip protein essential for social motility and colonization of tsetse fly salivary glands

We previously identified the cAMP response protein CARP3 (TriTrypDB entry Tb927.7.5340) in an RNAi screen for cAMP resistance in bloodstream forms (BSFs) of *T. brucei*[32]. CARP3 is a protein with no sequence homology outside *Trypanosoma* and unknown biochemical functions, except N-terminal myristoylation site(s) (http://lipid.biocuckoo.org/webserver.php). Its novel predicted structure (AlphaFold2[39]) is shown in Supplementary Fig. 1a. As cAMP regulates social motility (SoMo) in procyclic forms (PCFs)[8,40], we asked whether CARP3 did also control the parasite's colonization of the tsetse fly vector. A homozygous *CARP3* deletion mutant, its endogenous single allele *CARP3* rescue, as well as a tetracycline-inducible *CARP3* RNAi cell line were generated in the fully differentiation-competent pleomorphic *T. brucei* strain AnTat 1.1 'Munich'[41] and analyzed for in vitro culture phenotypes in SoMo, single cell motility, growth and differentiation. Deletion or depletion of *CARP3* caused a complete block in SoMo (Fig. 1a, b). In contrast, single cell mean velocity in viscous medium, growth of freshly differentiated PCFs and BSFs and development of slender to stumpy BSFs and further to PCFs were all unaffected (Supplementary Fig. 1b–g). The journey of trypanosomes in the tsetse fly vector proceeds through additional well-defined developmental stages. The parasites migrate from the insect's midgut via the cardia (proventriculus) to the salivary glands (SGs) (Fig. 1c). Upon ingestion of stumpy BSFs by tsetse flies, *carp3* KO cells were unable to colonize the SGs, while showing high midgut infection rates (Fig. 1d). The single-allele *CARP3* rescue was sufficient to restore SG infection rates to wild type levels. Indirect immunofluorescence analysis using a CARP3-specific polyclonal antibody revealed localization of CARP3 to the flagellar tip in PCFs (Fig. 1e, Supplementary Fig. 1h), consistent with subcellular proteome studies[16,17,42]. CARP3 is concentrated at the tip of the parental and daughter flagella (Fig. 1e) and it appears close to the flagellar membrane as a horseshoe-shaped signal with a slight anterior to posterior gradient (Supplementary Fig. 1h). No signal was detected in the *carp3* KO cell line (Fig. 1f). Surprisingly, in BSFs, CARP3 localized along the entire length of the flagellum and at the posterior cell pole (Fig. 1g, Supplementary Fig. 1i) with subcellular redistribution within the first hours of BSF to PCF differentiation. This dynamic localization of CARP3 during parasite development may correlate with life-cycle stage-specific functions. In PCFs, immunofluorescence analysis also suggested colocalization of CARP3 and Ty1-tagged adenylate cyclase isoforms ACP1 or ACP6 at the flagellar tip (Fig. 2a).

### Super-resolved spatial correlation of CARP3 with components of SoMo signaling

To precisely localize components of cAMP signaling involved in SoMo in the tip microdomain, we used photo-activated localization microscopy (PALM) for CARP3, ACP1 and phosphodiesterase PDEB1. One allele of *CARP3* was C-terminally fused in situ to photo-activatable mCherry (CARP3-PAmCherry). The fusion protein was fully functional in SoMo, as deletion of the wild type *CARP3* allele in the transgenic line did not impair SoMo on agarose plates (Fig. 2b). Quantitative colocalization analysis at single molecule resolution was used to detect distinct subdomains at the flagellar tip (radius ~300 nm). This confirmed close proximity of CARP3 to the flagellar tip membrane (Fig. 2c). In the same cell line, ACP1 was C-terminally fused in situ to mNeonGreen (ACP1-mNG) (Supplementary Fig. 2a). ACP1-mNG localized to the flagellar tip membrane (Fig. 2c and Supplementary Fig. 2b) as expected from the presence of a transmembrane domain (TMD) and consistent

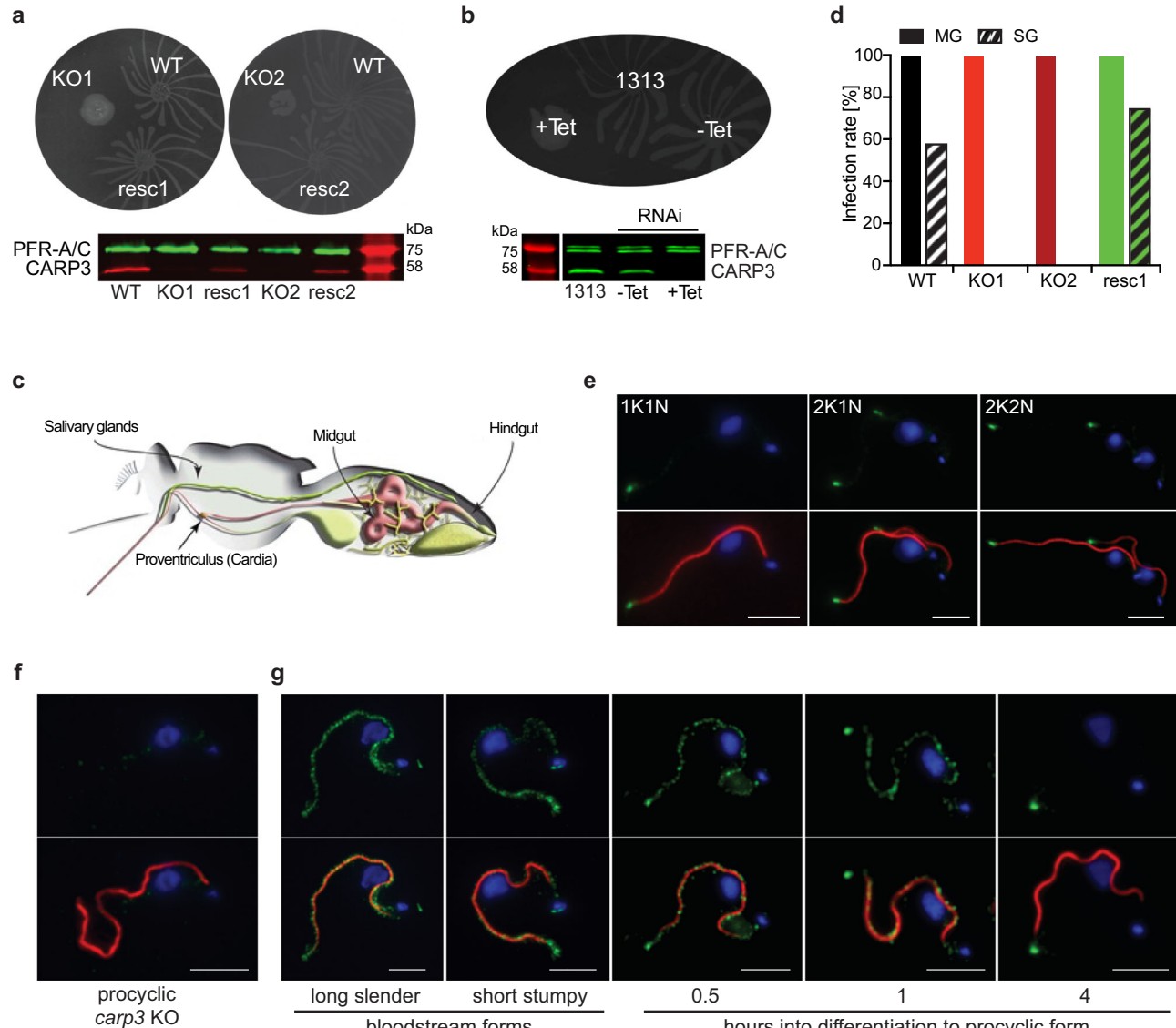

**Fig. 1 | CARP3 is a flagellar tip regulator of social motility (SoMo) and is essential for colonization of tsetse fly salivary glands. a** SoMo assays of procyclic *T. brucei* AnTat 1.1 wild type (WT), *carp3* knock-out (KO, independent clones KO1, KO2) or in situ *CARP3* rescue (resc, independent clones resc1, resc2). The Western blot was probed with anti-CARP3 and anti-PFR-A/C (loading control). **b** SoMo assay upon tetracycline (Tet)-inducible RNAi of *CARP3* (-Tet / +Tet 24 h) and the parental AnTat 1.1 1313 cell line. The Western blot shows CARP3 repression detected by antibodies as in (**a**). **c** Illustration of the digestive system and the salivary glands of a tsetse fly (adapted from[91]). **d** Infection rates of tsetse fly midgut (MG) or salivary glands (SG) with *T. brucei* AnTat 1.1 cell lines as in (**a**). Flies were dissected 34-36 days p.i., *n* (flies) = 48 (WT), 50 (KO1), 50 (KO2), 40 (resc1). 10 mM L-glutathione was included in the blood meal (Institute of Tropical Medicine Antwerp tsetse fly colony). Indirect immunofluorescence analysis of CARP3 (green) in *T. brucei* AnTat 1.1 procyclic form WT (**e**) or *carp3* KO (**f**). The upper panels show CARP3 (green) and the nuclear and mitochondrial DNA stained with DAPI (blue), the lower panels show an overlay with the axoneme (red; stained with the antibody mAB25). In (**e**) cells in different cell cycle stages are shown (1K1N, 2K1N, 2K2N; K kinetoplast, N nucleus). **g** Indirect immunofluorescence analysis of CARP3 (green) as in (**e**, **f**) during culture differentiation from bloodstream to procyclic forms. Scale bar in (**e–g**) 5 μm. Source data to (**a**, **b**) and (**d**) are provided as Source Data file.

with previous reports[17,18]. The probability of colocalization between two fluorescently tagged protein populations was determined by the coordinate-based colocalization (CBC) analysis method of Malkusch, et al.[43] (details see Methods). As positive control, we generated a cell line with one *CARP3* allele fused to PAmCherry and the other *CARP3* allele fused to mNeonGreen (Supplementary Fig. 2c). The CBC values' distribution for non-colocalizing proteins (negative control) was instead obtained by simulating two independent Poisson point patterns (Supplementary Figs. 2f, 3, see Methods for details). Comparison of the CBC values' distribution of CARP3-PAmCherry and ACP1-mNG (23.46% events with CBC ≥ 0.5) with the negative control (2.19% of events with CBC ≥ 0.5, see Supplementary Table 1) showed significant

colocalization for CARP3-ACP1 (*p* < 0.05) that was in the same range as the positive control (CARP3-PAmCherry-CARP3mNG; 22.22% events with CBC ≥ 0.5, see Supplementary Table 1). CBC values higher than 0.5 can be generally observed at the extreme tip of the flagellum (Fig. 2d–f).

PDEB1 was also C-terminally fused to mNeonGreen and expressed in the CARP3-PAmCherry line (Supplementary Fig. 2a). Live cell fluorescence microscopy shows localization of PDEB1-mNG along the flagellum in a tip to base decreasing gradient (Supplementary Fig. 2b). At single molecule resolution, PDEB1-mNG does not localize to the far anterior tip of the trypanosome flagellum, in agreement with previous localization to the paraflagellar rod (Fig. 2g)[8,30,42].

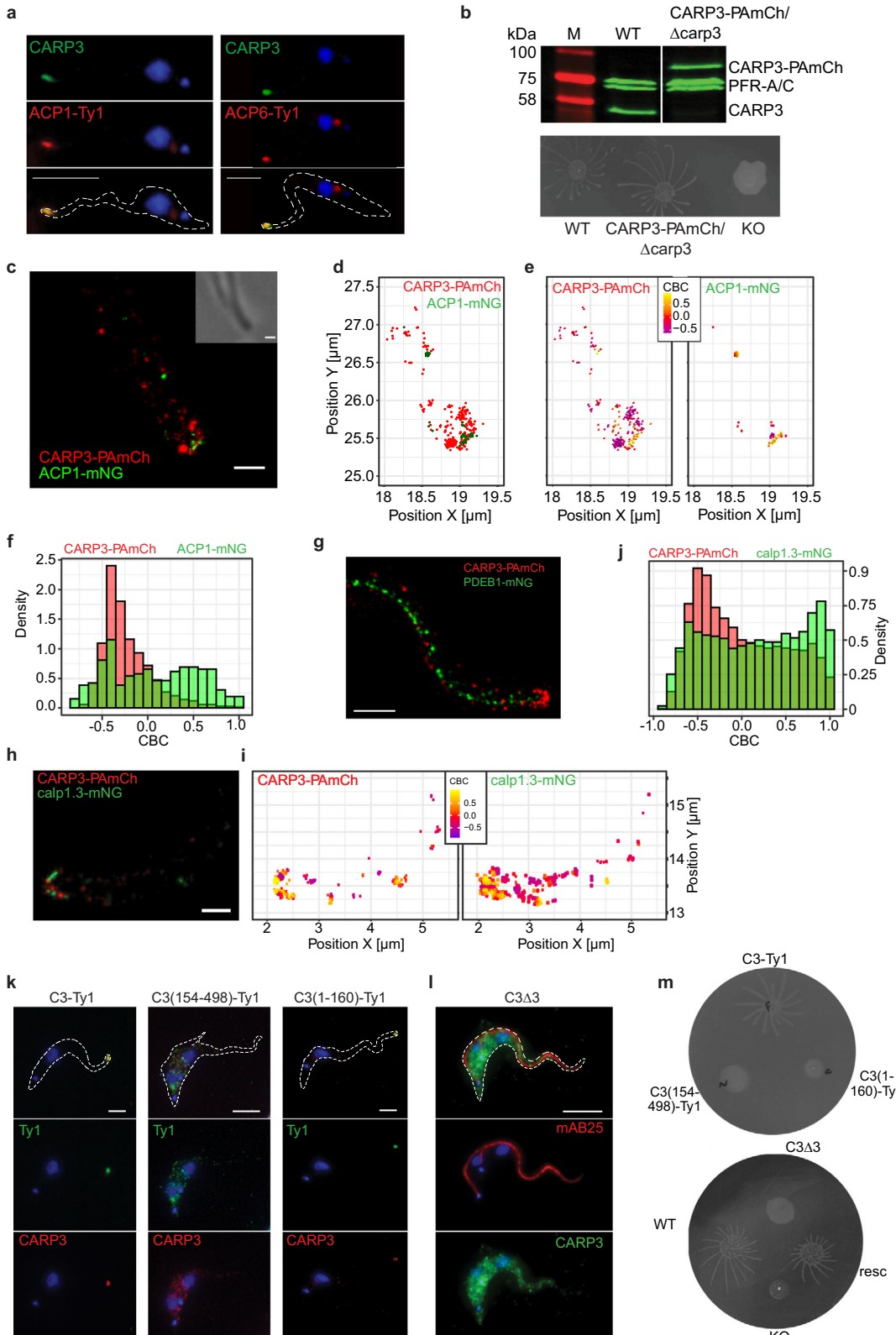

## The N-terminus of CARP3 is essential for flagellar tip membrane localization and SoMo

In absence of a predicted transmembrane domain, membrane-proximal localization of CARP3 suggests interaction with membrane(-associated) proteins or (dual) acylation[44,45]. In order to position CARP3 relative to the membrane at single-molecule resolution, we co-expressed CARP3-PAmCherry with an mNeonGreen fusion of the

previously characterized flagellar tip protein calpain 1.3 (Fig. 2h; Supplementary Fig. 2a) that is associated with the membrane via N-terminal dual acylation[14]. The CBC values' distribution for the CARP3-PAmCherry / calpain 1.3-mNG combination (Fig. 2i, j) showed significant colocalization with 33.9% of events with CBC ≥ 0.5, while the negative control remained at 2.19% ($p < 0.05$) and the positive control (CARP3-PAmCherry-CARP3mNG) at 22.22% (Supplementary Table 1

**Fig. 2 | Super-resolution topology in the flagellar tip microdomain. a** Indirect immunofluorescence analysis of CARP3 and ACP1-Ty1 (left) or ACP6-Ty1 (right) in procyclic *T. brucei* 29-13. Top: CARP3 (green) + DNA (blue, DAPI); middle: Ty1 (red) + DNA; bottom: merge. Scale bars 5 μm. **b** Western blot (top, probed with anti-CARP3 and anti-PFR-A/C as loading control) and SoMo assay (bottom) of procyclic *T. brucei* AnTat 1.1E CARP3-PAmCherry/Δcarp3 cells. Wild type (WT) and homozygous *carp3* knock-out (KO) serve as controls. M: molecular weight marker. **c** PALM imaging and (**d**) single molecule localization shown via centroids of CARP3-PAmCherry (PAmCh, red) and ACP1-mNeonGreen (mNG, green) at a procyclic *T. brucei* AnTat 1.1E fla-gellar tip. Corresponding phase contrast shown as inset in (**c**). Colocalization analysis of CARP3-PAmCherry with ACP1-mNeonGreen (**e**, **f**). CARP3 (**e**, left) and ACP1 (**e**, right) localizations are color-coded according to their respective Coordi-nate Based Colocalization (CBC) values, where a higher value signifies a higher colocalization probability. CBC histograms (**f**) of CARP3-PAmCherry (red) and ACP1-mNeonGreen (green) single molecule distributions. CBC values were calculated in a maximum radius of 300 nm and radius intervals of 5 nm. *n* = 19 flagella. **g** PALM imaging of CARP3-PAmCherry (red) and PDEB1-mNeonGreen (green) in a procyclic *T. brucei* AnTat 1.1E flagellum. PALM imaging (**h**) and colo-calization analysis (**i**, **j**) of CARP3-PAmCherry (red) and calpain 1.3-mNG (green) at a procyclic *T. brucei* AnTat 1.1E flagellar tip. CARP3 (i, left) and calpain 1.3 (i, right) localizations are color-coded according to their CBC values. Scale bars in (**c**), (**g**) and (**h**) 0.5 μm. Quantification in (**j**) shows CBC histograms of CARP3-PAmCherry (red) and calpain 1.3-mNG (green) single-molecule distributions. *n* = 15 flagella. **k, l** Indirect immunofluorescence analysis of CARP3 localization in procyclic *T. brucei* AnTat 1.1 upon constitutive overexpression of CARP3-Ty1, CARP3(154-489)-Ty1 or CARP3(1-160)-Ty1 or in situ add-back of *CARP3Δ3* in a *carp3* KO background. **k** Anti-CARP3 red, anti-Ty1 green, DAPI (DNA) blue. (**l**) mAB25 (anti-TbSAXO) red, anti-CARP3 green. Scale bars 5 μm. **m** SoMo assay of cell lines as in (**k, l**) including WT and *carp3* KO. Source data to (**b**) is provided as Source Data file.

and Supplementary Figs. 2e, f, 3). CARP3 was found in a chemical proteomic survey of myristoylated proteins[46] in addition to the sequence prediction of myristoylation. The structural model of CARP3, computed using AlphaFold2[39], shows a highly structured α-helical domain predicted with high confidence and interspaced by several long loops that are predicted with low confidence and likely represent intrinsically disordered (ID) regions (Supplementary Fig. 1a). To define the domain responsible for localization, deletion mutants of CARP3 were expressed in a *carp3* knock-out background (Supplementary Fig. 4a). Upon removal of the N-terminal 153 amino acids, C-terminally Ty1-tagged CARP3 was no longer localized at the flagellar tip, whereas expression of CARP3(1-160)-Ty1 was sufficient for flagellar tip targeting (Fig. 2k). CARP3(1-160)-Ty1 seems to be unstable as its expression level is too low for detection by Western blot (Supplementary Fig. 4a) but sufficient for detection by IFA (Fig. 2k). Upon deletion of the three N-terminal glycine residues (CARP3Δ3 mutant), no CARP3 was detec-ted at the flagellar tip (Fig. 2l). The expression level of CARP3Δ3 was comparable to that of full-length CARP3 (Supplementary Fig. 4b). The deletion analysis strongly supports an essential role of myristoylation as well as sufficiency of the N-terminal structured domain for tip localization. Strikingly, all cell lines devoid of CARP3 at the flagellar tip were deficient in SoMo (Fig. 2m). We thus conclude that the precise localization of CARP3 at the flagellar tip is critical for trypanosome SoMo. The SoMo deficiency of the C-terminal CARP3 deletion mutant (CARP3(1-160)-Ty1) is not informative due to very low expression (Fig. 2m; Supplementary Fig. 4a).

### FLAM8 is required for flagellar tip localization of CARP3
To define the extension of the CARP3 containing microdomain, colo-calization with FLAM8 (Flagellar Member 8) was tested. FLAM8 is a large cytoskeletal flagellar tip protein proposed to localize at the plus end of the axonemal microtubules in PCFs[15]. FLAM8 was C-terminally fused in situ to either YFP (FLAM8-YFP)[15] or mNeonGreen (FLAM8-mNG), respectively, and expressed in CARP3-mCherry or CARP3-PAmCherry cells (Supplementary Fig. 2a, b). Whereas wide field fluores-cence microscopy showed the expected tip localization of both CARP3-mCherry and FLAM8-YFP (Fig. 3a), at single-molecule resolu-tion, distinct zones for CARP3-PAmCherry and FLAM8-mNG were identified (Fig. 3c-f; Supplementary Fig. 5c). FLAM8 was found in the tip interior compared to the membrane-proximal CARP3 in flagellar longitudinal (Fig. 3c, d) and flagellar tip cross-sections (Fig. 3e, f). In a flagellar tip cross-section with a radius of approximately 300 nm, the two proteins were characterized by a colocalization interface in a zone ranging between 150 and 200 nm from the center (Supplementary Fig. 5c). The CBC values' distribution for the CARP3-PAmCherry / FLAM8-mNG combination shows 30.14% of events with CBC ≥ 0.5 compared to 2.19% for the negative control (Fig. 3g, Supplementary Fig. 2f, Supplementary Table 1). This significant difference ($p < 0.05$) is consistent with an association of a fraction of CARP3 with FLAM8 at the proposed localization of FLAM8 at the plus end of the axonemal microtubules[15]. Thus, separate pools of CARP3 seem to exist and FLAM8 might be involved in tip accumulation of CARP3. To test this hypothesis, CARP3-mCherry was expressed in the *flam8* homozygous deletion background[47]. Homogenous distributions of CARP3-mCherry (Fig. 3a) and endogenous CARP3 (Fig. 3b) were seen along the length of the flagellum as well as in the cytoplasm. In contrast, localization of FLAM8-YFP remained unchanged in a *carp3* knock-out cell line (Fig. 3a). Given the essential role of FLAM8 for the tip localization of CARP3, the SoMo deficiency of *flam8* KO cells was not surprising (Fig. 3h). Controls verified that *flam8* KO cells had single cell swimming velocity and growth properties comparable to the parental cell line (Supplementary Fig. 5a, b). Also, adding back a single *FLAM8* allele fully restored wild type SoMo (Fig. 3h). In summary, these data show that FLAM8 is essential for tip localization of CARP3 and that mislocaliza-tion of CARP3 is sufficient to produce the SoMo-deficient phenotype.

Proximity of CARP3 and FLAM8 was observed in all develop-mental stages of trypanosomes isolated from different tsetse fly ali-mentary tract compartments and salivary glands (Fig. 3i). The flagellar tip colocalization found in midgut (procyclic and long trypomasti-gotes) and cardiac stages (dividing and long epimastigotes) was con-trasting to the redistribution of both proteins along the length of the flagellum in short epimastigotes in the cardia as well as in attached epimastigotes and mammalian-infective metacyclic forms in the tsetse salivary glands. The CARP3 and FLAM8 localization pattern in salivary gland stages was similar to that in bloodstream forms (Fig. 1g, Sup-plementary Fig. 1i; FLAM8 in BSFs see[48]). The dynamic localization of FLAM8 and CARP3 in the life cycle suggests a distinct role at the fla-gellar tip in the parasite stages migrating in the fly's alimentary tract compared to localization along the flagellum in other stages.

### Transmission-deficient trypanosomes are developmentally competent
We then tested the ability of *carp3* and *flam8* KO parasites to colonize tsetse salivary glands and asked the crucial question whether the phenotypes were due to migration or to an intrinsic developmental defect. Both mutants were able to colonize the tsetse cardia, albeit with lower observed parasite densities and at a reduced rate as com-pared to wild type or parental control cells (Fig. 4a–d). Midgut infec-tion rates were barely affected for *carp3* KO (Figs. 1d, 4a) and reduced for *flam8* KO (Fig. 4c), respectively. The key finding is that not a single mutant trypanosome reached the salivary glands in any of the 355 flies dissected (100 flies infected with *carp3* KO, Fig. 1d; 255 flies infected with *flam8* KO, Fig. 4c) in contrast to wild type / control parasites or rescued lines. Thus, *CARP3* and *FLAM8* represent two genes that impact both SoMo and vector colonization when deleted.

To check the impact of *CARP3* deletion on the intrinsic develop-mental competence of the parasite, we differentiated *carp3* KO parasites in culture using inducible overexpression of the post-

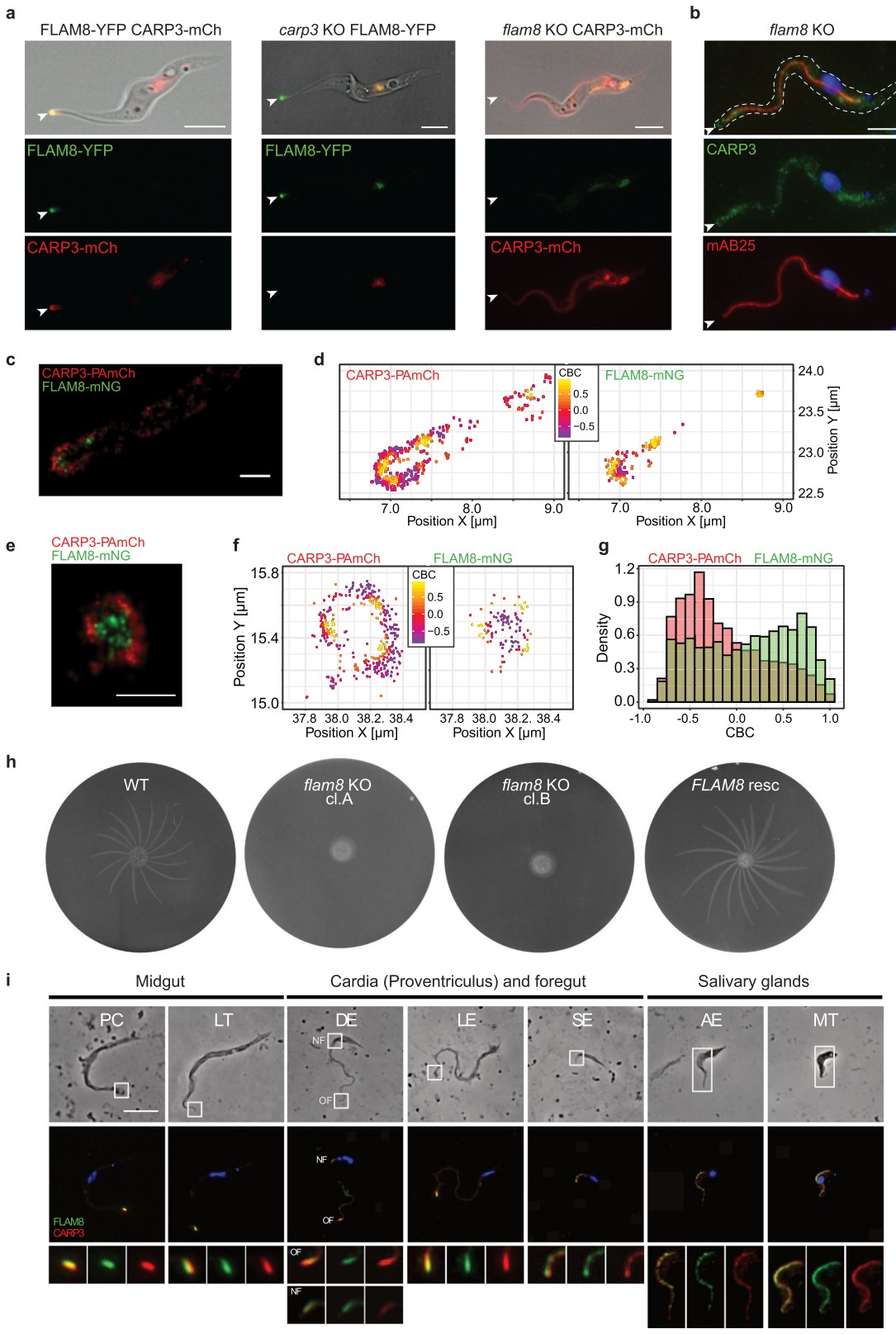

transcriptional regulator RBP6[49]. RBP6 overexpression was induced in wild type and *carp3* KO parasites (Supplementary Fig. 6), and followed by morphological analysis (Fig. 4e) and stage-specific marker expression profiling (Supplementary Fig. 6d, e). Efficiency and kinetics of differentiation from the procyclic form, via the epimastigote morphotype, to the metacyclic stage was identical for wild type cells and *carp3* KO cells. Similarly, development of *flam8* KO parasites was

analyzed in vivo by imaging the parasites at different time points during their cyclical development in the fly (Fig. 4f). Procyclic forms and long trypomastigotes (mesocyclics) were found in the tsetse posterior midgut, and long trypomastigotes and epimastigotes in the anterior midgut and cardia, showing that these developmental stages are formed and viable in the absence of FLAM8, hence in the absence of tip-localized CARP3. However, the number of long trypomastigotes

**Fig. 3 | FLAM8 is essential for flagellar tip localization of CARP3 and for SoMo.** **a** Live cell fluorescence microscopy of procyclic *T. brucei* AnTat 1.1E expressing FLAM8-YFP (green) and CARP3-mCherry (CARP3-mCh, red) (left), AnTat 1.1 *carp3* KO cells expressing FLAM8-YFP (center) or AnTat 1.1E *flam8* KO cells expressing CARP3-mCherry (right). Top: merge of fluorescent channels with DIC; middle: YFP; bottom: mCherry. **b** Indirect immunofluorescence analysis of CARP3 (green) in procyclic *T. brucei* AnTat 1.1E *flam8* KO cells. DNA was stained with DAPI (blue), the axoneme is labeled in red (detected by the antibody mAB25). Top: merge of fluorescent channels; middle: CARP3 + DAPI; bottom: mAB25+DAPI. White arrowheads in (**a**) and (**b**) point towards flagellar tips; scale bars 5 μm. PALM imaging (**c**) and colocalization analysis (**d**) of CARP3-PAmCherry (PAmCh) and FLAM8-mNeonGreen (mNG) at a procyclic *T. brucei* AnTat 1.1E flagellar tip. CARP3 (**d**, left) and FLAM8 (**d**, right) localizations are color-coded according to their respective CBC values. **e** PALM imaging of CARP3-PAmCherry (PAmCh) and FLAM8-

mNeonGreen (mNG) at a cross-section of a procyclic *T. brucei* flagellar tip. Scale bars in (**c**) and (**e**) are 0.5 μm. **f** Colocalization analysis as in (**d**) for a flagellar tip cross section. **g** CBC histograms of CARP3-PAmCherry (red) and FLAM8-mNG (green) single molecule distributions calculated from the data shown in (**c**–**f**). *n* = 27 flagella. **h** SoMo assay of procyclic *T. brucei* AnTat 1.1E wild type (WT), *flam8* KO (subclones A, B) or in situ *FLAM8* rescue (resc) cells expressing a fluorescent triple marker[92]. **i** Distribution of CARP3 and FLAM8 during the parasite cycle in the tsetse fly. Immunofluorescence on methanol-fixed trypanosomes obtained from dissected tsetse tissues four weeks after infection. Anti-CARP3 (red), anti-FLAM8 (green), DAPI (blue). Flagellum regions in white boxes on bright field are magnified in the bottom panel. Scale bar 5 μm. PC procyclic, LT long mesocyclic trypomastigote, DE dividing epimastigote, LE long epimastigote, SE short epimastigote, AE attached epimastigote, MT metacyclic, OF old flagellum, NF new flagellum.

in the anterior midgut and cardia was reduced in *flam8* KO parasites as compared to WT, suggesting that long trypomastigote parasites may be impaired in migration to or colonization of the anterior midgut and/or cardia. These results exclude a defect in the intrinsic developmental potential of the mutant trypanosomes and argue that the strong salivary gland colonization phenotype is specifically due to a sensory or signaling defect at the flagellar tip, possibly in long trypomastigotes, impinging on forward migration in the tsetse fly digestive tract.

## CARP3 interacts with ACs

As CARP3 was originally identified as a protein conferring cAMP resistance and is also linked to cAMP via the SoMo phenotype, we tested direct binding of cAMP or association with a cAMP-binding partner. Cyclic AMP coupled to agarose beads pulled down the cAMP-specific phosphodiesterase PDEB1-mNG (positive control) but not CARP3 from procyclic form lysates, indicating that CARP3 is not likely a cAMP-binding protein (Supplementary Fig. 7a). Its expression or localization in PCFs was also unaffected by even extreme intracellular cAMP concentrations elicited by the PDE inhibitors CpdA or CpdB[29] (Supplementary Fig. 7b). Such conditions completely inhibited SoMo (Supplementary Fig. 7c), as reported before[8].

Potential CARP3 interaction partners were then identified by GFP-trap pull-downs from *T. brucei* AnTat 1.1E CARP3-YFP of both BSF and PCF (Fig. 5a, b and Supplementary Fig. 8a, b). Mass spectrometry analysis identified CARP3 and 12 AC protein groups among the 16 protein groups significantly enriched ($p \leq 0.05$, $\geq 10$-fold enrichment) in BSFs (Fig. 5a, Supplementary data 1). In PCFs, CARP3 and 5 AC protein groups were among the 15 protein groups showing significant enrichment (Fig. 5b, Supplementary data 1). For independent confirmation, we selected ESAG4 (TriTrypDB entry Tb427.BES40.13), an abundant, BSF-specific AC, for which an antibody was available[33]. ESAG4 was pulled down by CARP3-YFP (Supplementary Fig. 8c) as well as by native CARP3 (Fig. 5c). For anti-CARP3-mediated pull-down, full-length ESAG4 was expressed as GFP fusion protein (cell line DNi-3 from Salmon et al.[33]) enabling its detection by in-gel fluorescence[50]. To identify the protein domain(s) of ESAG4 required for interaction with CARP3, the intracellular C-terminal tail (107 amino acids, ESAG4Δc107-GFP) following the AC catalytic domain of ESAG4 was truncated (cell line DNi-2[33]). This did not affect CARP3 interaction (Fig. 5c). In contrast, deletion of the AC catalytic domain (cell line ESAG4ΔCAT-GFP) abolished interaction. Both mutant proteins were expressed at a level comparable to full-length ESAG4-GFP (Fig. 5c, input). Mass spectrometry analysis of the reciprocal pull-down from ESAG4-GFP cells provided further confirmation of ESAG4-CARP3 interaction (Supplementary Fig. 8d; Supplementary data 2). Besides CARP3, other AC isoforms were enriched by ESAG4-GFP pull-down, indicating AC dimerization, as previously reported[18,27,33]. Consistently, a highly confident homo-dimeric structure was predicted using AlphaFold (Supplementary Fig. 9d; Supplementary data 7). The physical interaction between CARP3 and ACs was independently suggested by a CARP3 proximity proteomics approach in BSFs (CARP3

BioID; details see Supplementary Methods) that revealed a significant enrichment (3-115-fold, FDR ≤ 0.05, $s_0 = 2$) of several AC isoforms (Supplementary Fig. 8e; Supplementary data 3). In this proximity screen, we also found FLAM8 and calpain 1.3 enriched 125- and 83-fold, respectively; a result consistent with the high degree of colocalization of FLAM8, calpain 1.3 and CARP3 in PCFs. Thus, a core complex seems to be maintained at least throughout parts of the parasite life cycle. Physical interaction of CARP3 with PCF-specific flagellar tip-localized AC isoforms was directly shown by pull-down of ACP4-mNG and ACP5-mNG via CARP3 (Supplementary Fig. 8f, g). The CARP3-AC interaction was furthermore supported by structural modeling using AlphaFold[39] as well as the recently released AlphaFold multimer[51], which predicted models of CARP3 in complex with each of seven different AC isoforms (ESAG4, GRESAG4.1, ACP1, 3, 4, 5, 6). Superposition of all complex models shows a high degree of overall structural similarity (root-mean-square-deviation of 0.8–1.4 Å for >90% of all main-chain atoms) (Supplementary Fig. 9a-c, Supplementary data 6). The predicted interface consists of the AC catalytic core and two N-terminal helices (residues 13–41) of CARP3. The calculated buried surface area (600–700 Å²) is small, but the predicted local-distance difference test (pLDDT) score >90 and a low predicted alignment error (PAE) (<5 Å) provide a very high degree of model confidence. As negative control, structure prediction using AlphaFold for a receptor-type AC of the distantly related kinetoplastid *Bodo saltans* and *T. brucei* CARP3 resulted in a low confidence model with PAE > 20 Å for putative interface residues.

## CARP3 regulates AC abundance in trypanosomes

We expected the CARP3-AC interaction to modulate AC catalytic activity in the complex. CARP3 and ESAG4 were co-expressed in mammalian HEK293 cAMP reporter cells that express luciferase under control of a cAMP response element (CRE). The CARP3-AC interaction was maintained in the heterologous system as shown by pull-down (Fig. 5d). This provided additional evidence for direct physical interaction independent of the parasite cellular environment. Surprisingly, the amount of cAMP produced by recombinantly expressed ESAG4 and normalized to the ESAG4 level was independent of the amount of co-expressed CARP3 in the heterologous HEK cell system (Supplementary Fig. 8h). In contrast, *CARP3* knock down resulted in strongly reduced cAMP production (swell dialysis AC assay) (Fig. 6a) and a decrease in ESAG4 levels over 24 h in BSFs (Fig. 6b). Quantitative proteomics confirmed a 13-fold and ~2-4-fold decrease of CARP3 and five AC groups, respectively (Fig. 6c, Supplementary data 4). No other significant proteome changes were detected ($p \leq 0.05$). The proteomic data show that CARP3 is a multi-AC regulator (Supplementary data 4, 5). A 3-5-fold decrease in the expression of ACP1, 4 and 6 upon *CARP3* knock down determined by in-gel fluorescence and Western blotting in procyclic cells (Fig. 6d) confirmed the regulation in PCF and suggested that the SoMo and fly colonization phenotypes elicited by CARP3 are likely due to changes in AC abundance. Surprisingly, ACP5 levels increased 2.3-fold upon CARP3 repression. Taken together, these

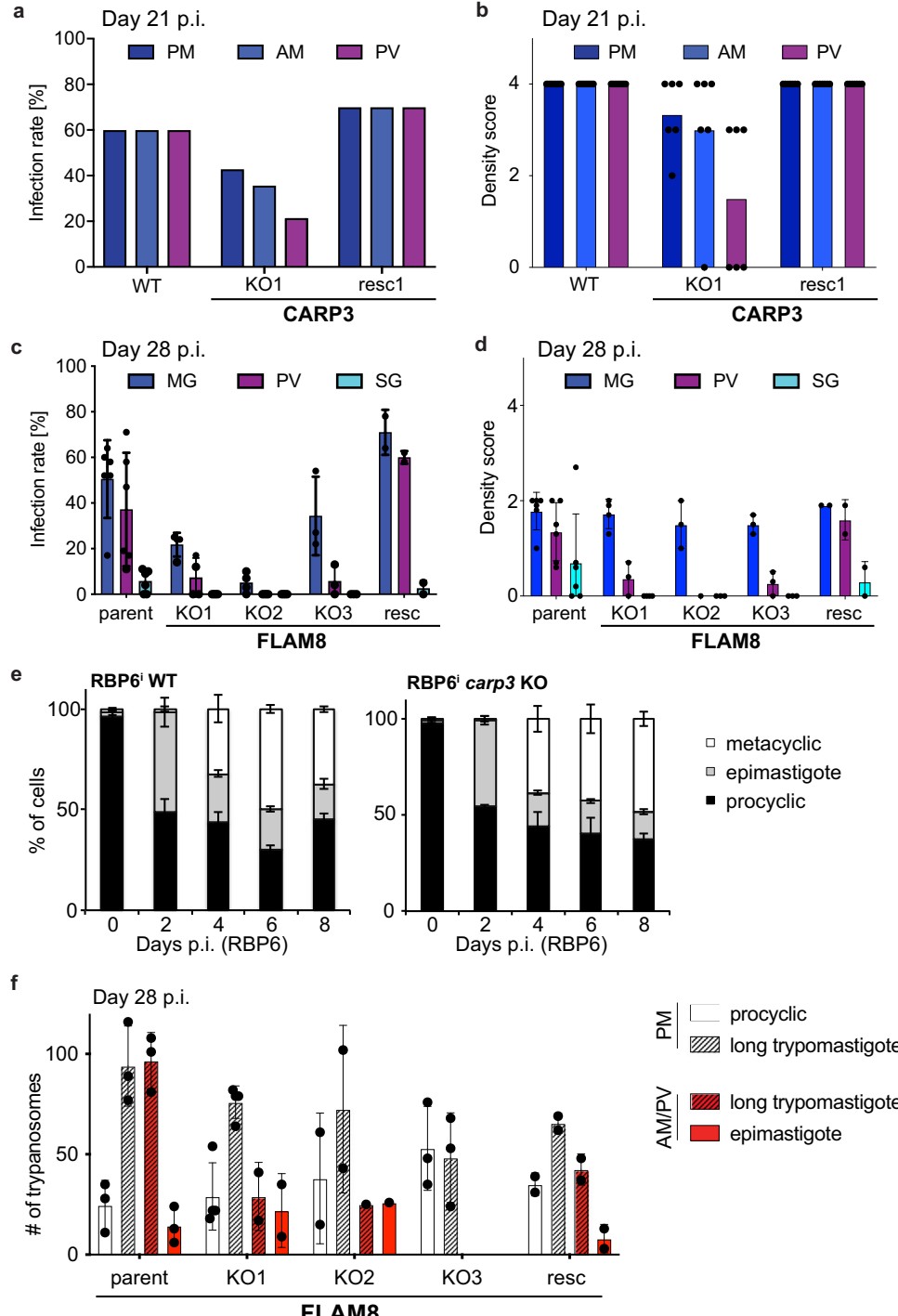

**Fig. 4 | Development and fly infection phenotypes of *carp3* and *flam8* KO.**
**a** Infection rates and (**b**) parasite densities in posterior midgut (PM), anterior midgut (AM) or proventriculus (PV or cardia) of tsetse flies infected with *T. brucei* AnTat 1.1 WT, *carp3* KO1 or in situ *CARP3* rescue (resc1) cells (Institute of Tropical Medicine Antwerp fly colony). Flies were dissected 21 days p.i. *n* (flies) = 15 (WT), 14 (KO1), 10 (resc1). Parasite densities were scored only in flies with positive midgut infection. Density scoring (parasites per field): '4' > 1000, '3' 100-1000, '2' 10-100, '1' 1-10, '0' no parasites. Note that in contrast to Fig. 1d no L-glutathione was added to the bloodmeal, resulting in lower midgut infection rates but increased sensitivity of phenotype detection. **c** Infection rates and (**d**) parasite densities in MG, PV or salivary glands (SG) of tsetse flies infected with *T. brucei* AnTat 1.1E 'Paris' (parent), *flam8* KO (subclones KO1, KO2, KO3) or in situ *FLAM8* rescue cells (Institut Pasteur Paris fly colony). All cell lines express a red fluorescent triple marker[92]. Flies were dissected 28 days p.i.; Mean ± SD of *n* (replicates, flies) = 6, 122 (WT); 4, 88 (KO1); 4,

109 (KO2); 3, 58 (KO3); 2, 40 (resc). Density scoring as in (**b**). Absolute transmission efficiencies differ between CARP3 and FLAM8 experiments due to different fly colonies and parasite strains. **e** RBP6-induced development in *carp3* KO and parental RBP6 line. Procyclic, epimastigote and metacyclic forms were classified according to morphology, kinetoplast position relative to nucleus (DAPI staining) and expression of stage-specific marker proteins (see Supplementary Fig. 6). >100 cells were analyzed at each time point and replicate. Mean ± SD of *n* = 3.
**f** Quantification of different developmental stages (procyclic, long mesocyclic trypomastigote, epimastigote) in PM or AM/PV of tsetse flies infected with *T. brucei* AnTat 1.1E 'Paris' (parent), *flam8* KO (subclones KO1, KO2, KO3) or *FLAM8* rescue cell lines. Flies were dissected 28 days p.i.; *n* (flies, trypanosomes) = 3, 689 (parent); 4, 522 (KO1); 2, 272 (KO2); 3, 304 (KO3); 2, 302 (rescue). Source data to (**a**–**f**) are provided as Source Data file.

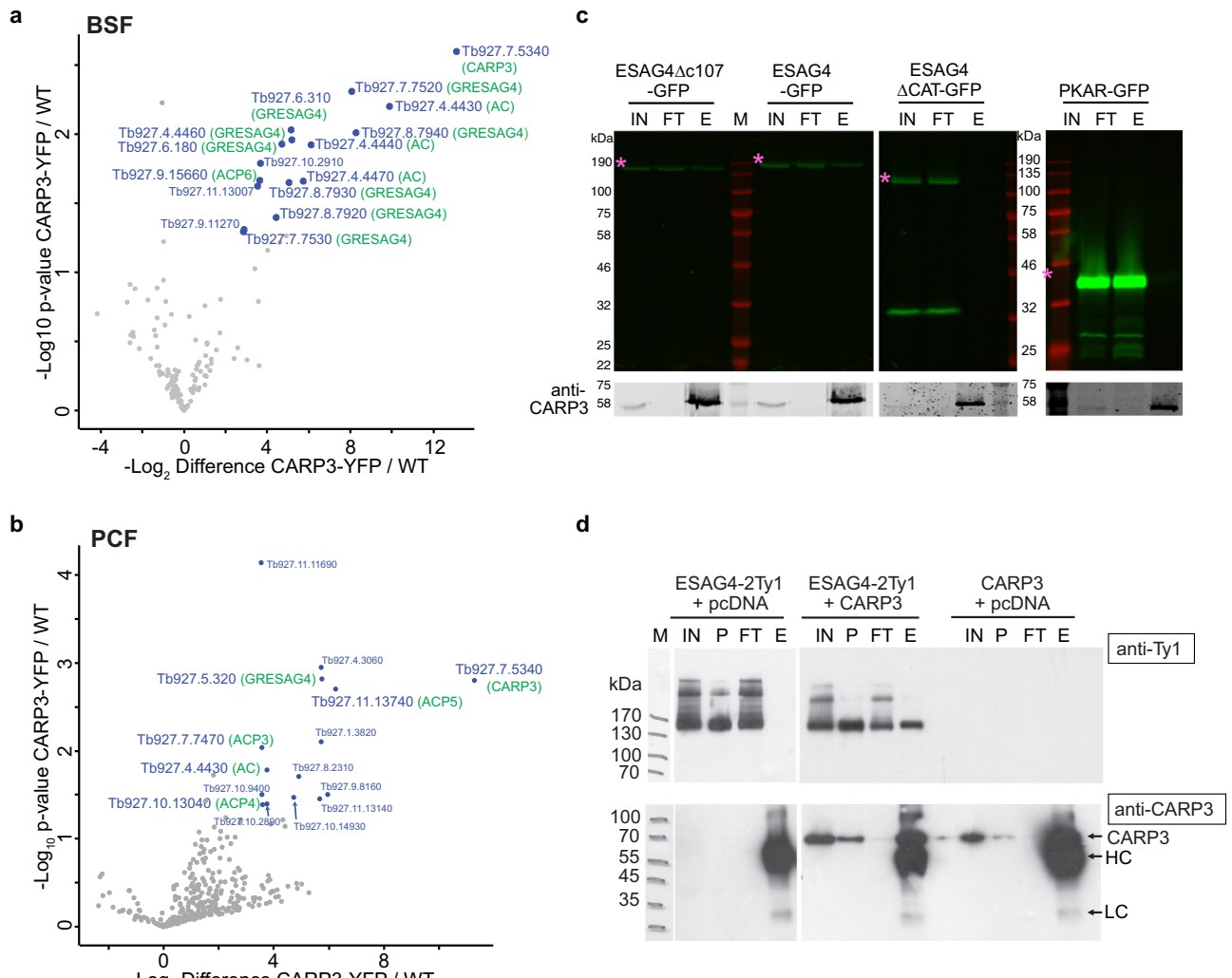

**Fig. 5 | CARP3 interacts with ACs.** Volcano plot representation of proteins identified upon GFP trap pull-down in *T. brucei* AnTat 1.1E CARP3-YFP cells compared to wild type (WT) in BSFs (**a**) or PCFs (**b**) (*n* = 2 replicate pull-downs each). Identified proteins are plotted according to *p* value and fold change with significantly enriched proteins (*p* value ≤0.05, fold change ≥10) shown in blue with their TriTrypDB entries (https://tritrypdb.org). The bait protein CARP3 and AC isoforms are indicated in green. **c** CARP3 immunoprecipitation (anti-CARP3 coupled to protein A beads) in BSF cell lines expressing ESAG4-GFP, ESAG4Δc107-GFP, or ESAG4ΔCAT-GFP, respectively. Pull-down assays in a PKAR-GFP cell line served as negative control. Upper panels: detection of GFP by in-gel fluorescence; lower panels:

Western blot detection of CARP3. The asterisk labels the fluorescent band corresponding to the respective GFP fusion protein. IN input; FT flow-through; E elution (10x load of input or flow-through); M protein molecular weight marker. **d** CARP3 immunoprecipitation (anti-CARP3 coupled to protein A beads) from soluble fractions of HEK293 cells expressing CARP3 or ESAG4-2Ty1 or both. Equal amounts of total DNA were transfected in all conditions (ratio of transfected plasmids 1:1). The upper Western blot was probed with anti-Ty1, the lower with anti-CARP3. IN soluble input, P insoluble pellet, FT flow-through, E elution, M protein molecular weight marker, HC anti-CARP3 heavy chain, LC anti-CARP3 light chain. Source data to (**c**) is provided as Source Data file.

results show that CARP3 is a novel modulator of AC abundance in the flagellar tip cAMP microdomain.

## Discussion

In this work, we propose a novel architecture of a cAMP signaling complex essential for successful arthropod host-parasite interaction and hence transmission of trypanosomes. CARP3, a protein with no sequence homologue outside the genus *Trypanosoma*, interacts with and regulates members of the receptor-type AC family. The model of the complex at the flagellar tip membrane as illustrated in Fig. 6e, is based on orthogonal evidence from pull downs and super-resolution microscopy in PCF as well as AlphaFold structure predictions of CARP3 in complex with PCF-specific AC isoforms. CARP3-ACP3 and −5 interaction was also recently shown by Shaw, et al.[52] and a proximity proteomics study in PCF identified CARP3 as putative ACP1 interactor[17]. Phenotypes upon expression of deletion mutants of the ESAG4 cyclase

are in full agreement with the high confidence prediction of an interface between the conserved AC catalytic core domain and two N-terminal helices of CARP3. The very similar structure predictions of seven different AC isoforms in complex with CARP3 (Supplementary Fig. 9a, b, Supplementary data 6) suggest that CARP3 may be a pan-cyclase interactor in trypanosomes. This is in agreement with identification of members of several AC protein clusters in CARP3-YFP pull-downs. The N-terminal myristoylation of CARP3, revealed by the CARP3Δ3 mutant and previous chemical proteomics data[46], is essential for tip localization and may enhance AC complex formation by transient membrane interaction of CARP3. N-myristoylation confers transient membrane association to proteins, but membrane localization generally needs to be stabilized by a second mechanism, either an interaction with a membrane protein, subsequent palmitoylation or by ionic interactions of basic surface residues with the membrane[53,54]. CARP3 has no palmitoylatable cysteine at position three and has not

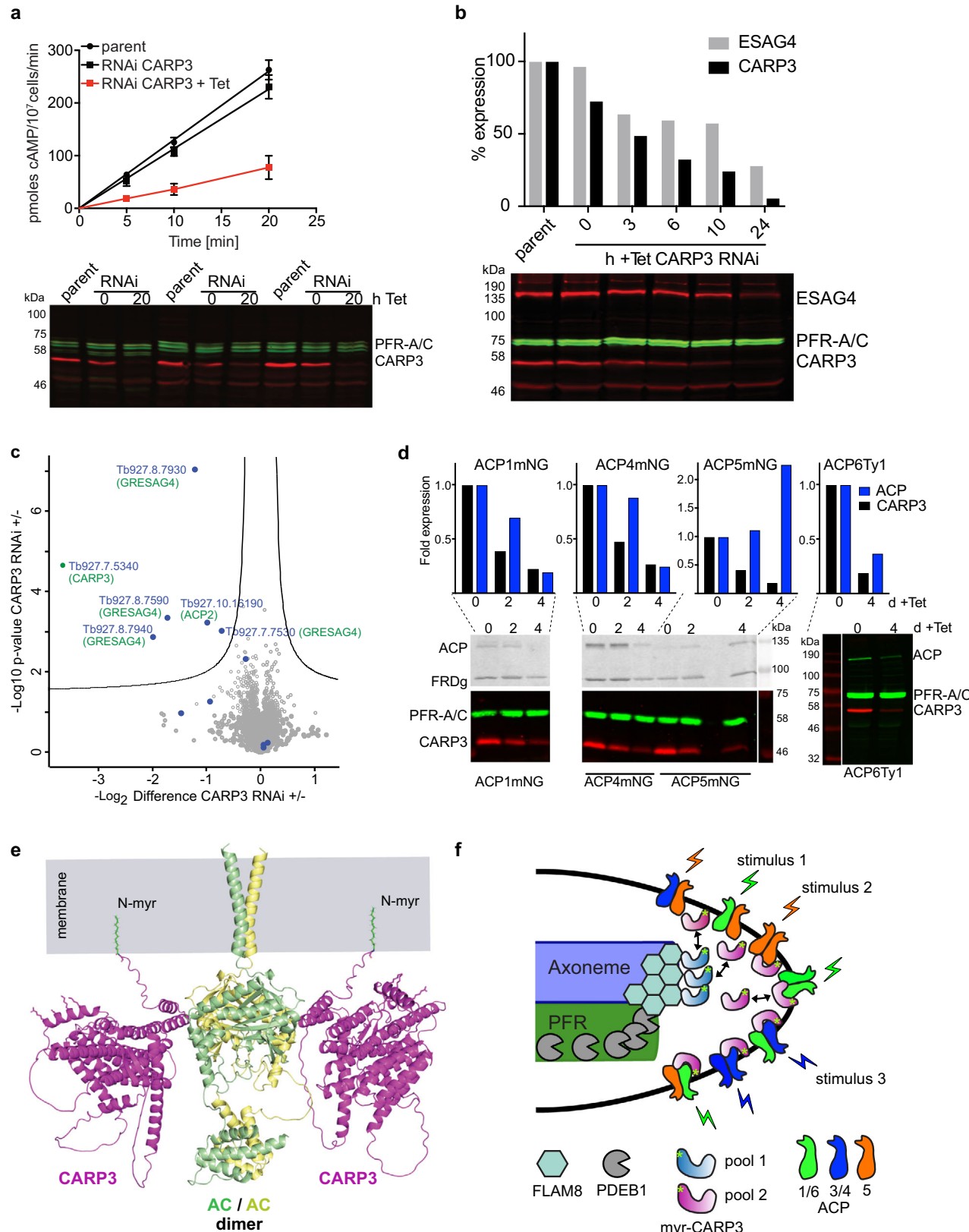

been identified in a palmitoyl proteome[55]. The structure model allows membrane contact of CARP3 by a helix rich in basic residues and membrane insertion of a myristoyl group attached to CARP3 while in complex with ACs (Fig. 6e). Thus, a cooperative effect of AC-interaction, N-myristoylation and basic surface residues is possible. It is conceivable that calpain 1.3 is also part of the same membrane

microdomain. Although we have no evidence for a direct physical interaction with the AC-CARP3 complex, PALM microscopy shows extensive colocalization of calpain 1.3 with CARP3 at single molecule resolution at the procyclic flagellar tip, and BioID with CARP3 as bait identifies calpain 1.3 (83-fold enriched) in BSF. As calpains are linked to $Ca^{2+}$ signaling in other systems, calpain 1.3 may be involved in $Ca^{2+}$-

**Fig. 6 | CARP3 regulates AC abundance in BSFs and PCFs. a** Total AC activity in *T. brucei* BSF MiTat 1.2 13-90 (parent) or *CARP3* RNAi ± 1 µg/mL Tet, 20 h. Mean ± SD of *n* = 9 (3 independent replicates controlled by Western blot with PFR-A/C as loading control). **b** Western blot analysis of CARP3 and ESAG4 in *T. brucei* BSF MiTat 1.2 13-90 (parent) or Tet-induced *CARP3* RNAi over 24 h. Signals were normalized to PFR-A/C, parent was set to 100%. **c** Volcano plot of proteins identified by label-free proteomics in *T. brucei* BSF MiTat 1.2 13-90 upon *CARP3* RNAi ± Tet for 24 h. Proteins are plotted according to *p* value and fold change. AC isoforms are labeled as blue dots. Protein IDs that differ significantly (two-sided Student's *t* test, *p* value ≤0.05, $s_0$ = 0.1) in abundance are located above the significance line with their TriTrypDB entries (https://tritrypdb.org) and gene names (in green) indicated. **d** Analysis of ACP abundance by in-gel fluorescence (C-terminally mNeonGreen tagged ACP1/4/5) or Western blot (C-terminally Ty1 tagged ACP6) in PCF trypanosomes upon *CARP3* RNAi ± Tet. CARP3 (anti-CARP3) and ACP signals were normalized to PRF-A/C and day 0 was set to 1; FRDg as loading control. **e** Cartoon model of the CARP3-AC complex at the membrane including the transmembrane helix and the intracellular parts of an AC dimer (ESAG4). CARP3 N-myristoylation (N-myr) is indicated. The side chains of several basic amino acids of a membrane-proximal helix predicted for CARP3 are shown. A second CARP3 molecule is shown here, as the AC dimer model sterically allows interaction with two CARP3 molecules. **f** Model illustrating the relative localization of CARP3, AC, FLAM8 and PDEB1 at the anterior tip of the procyclic trypanosome flagellum. Two pools of CARP3 are indicated based on PALM. AC homo- and heterodimerization is indicated by different colors with flagellar tip ACs grouped and colored according to their role in SoMo: ACP1/6 (green) inhibit SoMo, ACP5 (orange) promotes SoMo and ACP3/4 (blue) were not linked to SoMo. Source data to (**a**, **b**) and (**d**) are provided as Source Data file.

regulation of trypanosome ACs, a property that has been reported earlier for some trypanosome ACs[56–59]. Calpain 1.3 differs from classical calpains by lacking some of the features critical for catalytic Ca²⁺-dependent protease activity[14]. CARP3 may serve an anchoring role for recruitment of additional proteins in AC signaling complexes. Several significant hits in the BioID interaction screen and the presence of intrinsically disordered regions (IDR loops) on the cytoplasmic face of the CARP3 structure model (Fig. 6e) are compatible with this interpretation. IDRs often provide a platform for multiple weak interactions in signaling complexes and are more likely to phase separate. An example is the membrane associated pLAT-Grb2-Sos1 condensate in T-cell receptor activation[60]. A fraction of CARP3 at the flagellar tip does not localize at the membrane but colocalizes with the cytoskeletal protein FLAM8, previously positioned at the plus end of the axonemal microtubules in PCF[15] and also distributed along the flagellum in BSF[48]. The close proximity and likely interaction of CARP3 and FLAM8 is supported by BioID in BSF (125-fold enrichment of FLAM8 with two biotinylation sites identified). Flagellar tip localization of CARP3 is dependent on FLAM8; in its absence, CARP3 still entered the flagellum but was not enriched at the tip. In the short epimastigote stage, CARP3 also followed the FLAM8 redistribution along the length of the flagellum[48]. FLAM8 therefore seems to be required for flagellar transport of CARP3 or unloading or concentration at the tip in PCF, possibly to increase local concentration. The intraflagellar transport (IFT) machinery is required for flagellar tip localization of FLAM8[61]. The complex of CARP3 with AC isoforms was detected in BSFs and PCFs. The proximity of CARP3 to FLAM8 and calpain 1.3, supported by super-resolution microscopy in PCF and by BioID in BSF, suggests that similar core complexes are present in several stages of the parasite's life cycle. So far, only phenotypes dependent on localization at the procyclic flagellar tip have been observed. In contrast to the aforementioned interactions, PDEB1 is not enriched in the tip microdomain but forms a gradient with a higher concentration close to the flagellar tip and a lower concentration at the flagellar base (Supplementary Fig. 2b). This indicates a diffusion barrier function to establish a cAMP microdomain at the tip in analogy to comprehensive studies in the mammalian system[23]. The previously described SoMo-negative phenotype of *pdeb1* KO parasites[9] is therefore likely due to disruption of a flagellar cAMP gradient.

*CARP3* deletion completely blocks trypanosome SoMo (Fig. 1 and[52]), and interestingly, the absence of CARP3 from the flagellar tip (in *flam8* KO and *carp3Δ3*) is alone sufficient for this phenotype. SoMo phenotypes produced by mutant CARP3, ACs[7,52] and PDEB1[8,9] together provide strong evidence that cAMP signaling is controlling SoMo. The same pathway has been suggested to also control the parasite's ability to orient and successfully migrate in the insect digestive tract to ultimately reach the salivary glands and complete the infectious cycle[9,62]. Analysis of our *carp3* and *flam8* KO parasites provides the currently highest degree of phenotypic specificity and support for a common signaling mechanism underlying SoMo and salivary gland colonization. (1) Both *carp3* and *flam8* KO cell lines do not show any growth phenotypes in culture and their swimming velocity is unchanged compared to controls; (2) *carp3* KO cells can proceed through complete differentiation in the RBP6-induced culture model with wild type level of metacyclogenesis, thus excluding a cell autonomous defect of the developmental program. Furthermore, the *flam8* KO cell line that shows mislocalized CARP3 developed into epimastigote stages in vivo in the tsetse alimentary tract. (3) In contrast to previous studies[9,52], clones of the fly transmissible pleomorphic AnTat 1.1 strain and freshly differentiated procyclic forms of this strain were used. Fly transmissions were done in two different laboratories with several independent KO lines and rescue lines. (4) Fly infection phenotypes are very similar for *carp3* and *flam8* KO parasites, although the absolute infection rate differs between the AnTat 1.1 clones, fly colonies and protocols in the two laboratories. Not a single trypanosome reached the salivary glands, even in flies with positive cardia infection. The observed slightly reduced midgut and reduced cardia infection rates may indicate an impact of the *carp3* and *flam8* KO mutants on SoMo at several steps of the parasite's journey through the insect. The tsetse infection phenotype of the *carp3* KO parasite strain used by Shaw, et al.[52] is very different. They observed strongly reduced midgut infection rates that contrast with high midgut infection rates of our *carp3* KO strain that specifically showed compromised proventriculus infection and absence from salivary glands. The *pdeb1* KO mutant[9] was not able to colonize the ectoperitropic space and could not establish a cardia infection. This may indicate involvement of PDEB1 in additional earlier steps of development or parasite fitness in the tsetse fly. The inherent fly transmission defect of the laboratory-adapted monomorphic Lister 427 strain[63–65] used by Shaw, et al.[9,52] complicates interpretation of the *pdeb1* KO phenotype.

Interestingly, in long trypomastigotes and long dividing epimastigotes colonizing the anterior midgut, cardia and foregut, i.e. in the migrating stages, FLAM8 is progressively enriched at the flagellar tip microdomain (Fig. 3i and[48]), where CARP3-AC complexes are required. FLAM8 and consequently CARP3 are redistributed from the tip to the length of the flagellum in short epimastigotes, a life cycle stage that normally attaches to the salivary gland epithelium via the flagellar membrane[66]. The change in subcellular localization may be cause or consequence of this attachment, but the salivary gland colonization defect suggests a role of the tip signaling complex prior to attachment. Together, *carp3* and *flam8* KO phenotypes provide the best available evidence that an AC signaling complex controls a crucial process leading to the colonization of fly salivary glands and thus parasite transmission. In this context, the SoMo phenotype in vitro, controlled by the same signaling proteins, serves as proxy for this complex process of migration of trypanosomes through the different anatomical regions of the fly in vivo. Chemotaxis seems to be an obvious link between the projections on agarose plates and the directional migration in the tsetse alimentary tract. Chemotaxis has been described for trypanosomes in culture[67] and a recent study demonstrated pH taxis of

trypanosomes undergoing SoMo on agarose plates[52]. However, major determinants of the collective swimming seen in the ectoperitrophic space of the gut[11] might also be dimensions and curvature of the confinement in that environment, as was recently demonstrated for *Chlamydomonas* algae[68]. There is currently no evidence that collective swimming in vivo and SoMo on plates are mechanistically linked, whereas directional migration and SoMo share dependence on tip AC complex signaling.

The proteins known to be involved in SoMo all impact on (local) cAMP concentration; ACs produce, PDEs degrade cAMP and CARP3 confers sensitivity to the toxic effect of PDE inhibitors[32], probably by stabilizing ACs, as shown in Fig. 6a-d. It is therefore reasonable to assume that the SoMo phenotype is due to changes in flagellar tip cAMP concentration [cAMP$^{tip}$]. To integrate the available data into a consistent model, we propose to consider AC dimerization (Fig. 6e, Supplementary Fig. 9d)[56,69] that has also been shown in vivo[18,31,33] (Fig. 6f). It seems contradictory that *carp3* KO cells and the sKO of *ACP5*[52] (but not ACP5$^{RNAi}$)[7], both expected to reduce [cAMP$^{tip}$], abolish SoMo, while RNAi-mediated depletion of ACP1 or ACP6 (also expected to reduce [cAMP$^{tip}$]) has apparently the opposite effect[7,52]. Different trypanosome AC isoforms may have different intrinsic cyclase activities and may be activated or inhibited by different stimuli. Upon heterodimerization, this may result in activation or dominant-negative inhibition, dependent on AC isoform composition of the tip domain and changing environmental stimuli to fine-tune cAMP levels. Dominant-negative heterodimer formation has previously been described for invertebrate soluble guanylate cyclase (GC)[70], mammalian AC8 in vascular smooth muscle cells[71] and also ESAG4 of trypanosomes[33]. In agreement with the model, AC isoform diversity and non-redundancy at the flagellar tip[7] and differential expression of transcripts encoding AC isoforms in trypanosomes colonizing midgut, cardia or salivary glands[36,37] have been reported. CARP3 depletion might result in compositional imbalance of tip AC isoforms. This is supported by the differential regulation of ACP1, 4, 6 (down) and ACP5 (up) upon CARP3 depletion. It is conceivable that AC complex signaling is required at several locations and bottlenecks in the fly alimentary tract and that different AC isoforms respond to different environmental cues. The trypanosomal receptor-type ACs have been hypothesized to be activated by binding of ligands to their extracellular part containing two VFT (Venus fly trap) domains[27,72] homologous to ligand-binding surface proteins in other systems[73].

Our model implies that future research should first focus on the AC family to unravel the cAMP signaling initiated at the flagellar tip. Detailed biochemical and structural analyses are required to identify AC-activating ligands and better understand the mechanism of cyclase activation, the role of homo- and heterodimerization and the unique isoform diversity and stage regulation of the trypanosomal AC family. Using the SoMo phenotype as simple in vitro assay, a key signaling system coordinating host-parasite communication and securing successful transmission of the parasite becomes accessible.

## Methods

### Trypanosome culture conditions

Bloodstream forms of the pleomorphic *Trypanosoma brucei brucei* strains AnTat 1.1 'Munich' or AnTat 1.1E 'Paris'[41] were cultivated at 37 °C and 5% CO$_2$ in modified HMI-9 medium[74] supplemented with 10% (v/v) heat-inactivated fetal bovine serum (FBS) and 1.1% methylcellulose (only for AnTat 1.1 'Munich'). Cell density was monitored using a haemocytometer and was kept below $8 \times 10^5$ cells/mL for continuous growth of replicative long slender bloodstream forms. Procyclic stage cells were generated by density-dependent differentiation of long slender bloodstream forms to growth-arrested short stumpy bloodstream forms (culture with starting density of $5 \times 10^5$ cells/mL was grown for 36 h without dilution). Short stumpy forms were transferred into modified DTM medium[75] complemented with 15% (v/v) heat-

inactivated FBS at $2 \times 10^6$/mL, followed by addition of 6 mM cis-aconitate and cultivation at 27 °C. Stumpy forms of AnTat1.1 'Paris' were generated by 48 h incubation in 5μM 8-pCPT-2'-O-Me-5'-AMP (BIOLOG Life Science Institute (Germany)). Procyclic forms of AnTat 1.1 'Munich', AnTat 1.1E 'Paris', AnTat 1.1 EATRO1125 T7T or MiTat 1.2 29-13[76] were grown at 27 °C in SDM-79 medium[77] supplemented with 10% (v/v) heat-inactivated FBS and 20 mM glycerol.

### Photoactivated localization microscopy

$1.6 \times 10^7$ trypanosomes were fixed in 2% PFA for 20 min at room temperature, washed 2-4 times with PBS/10 mM glycine and resuspended in 50 μL PBS. 5 μL fixed cells were mixed with 1.5 μL (1:5000) 100 nm diameter fluorescent TetraSpeck Microbeads (Thermo Fisher Scientific) and 200 μL PBS and loaded into an 8 well glass bottom μ-slide (Ibidi, Martinsried, Germany). Slides were centrifuged for 10 min at $1400 \times g$ and imaged on a Zeiss Elyra P.1 microscope (Carl Zeiss Microscopy GmbH, Jena, Germany) equipped with an Andor EM-CCD iXon DU 897 camera.

PAmCherry was activated by an HR diode 50 mW 405 nm laser (Power linearly increased over the course of the experiment and dependent on the protein abundance) and excited via a 200 mW 561 nm OPSL laser line (15% power) (Emission filter: LP 570). The 405 line trigger was set to "none" (continuous conversion) while the 561 line trigger was set to "integration" (laser active only when the camera is collecting photons). MNeonGreen was excited via a 200 mW 488 nm OPSL laser line (15%) (no conversion required) (Emission filter: BP 495–590 + LP 750) (laser trigger set to "integration").

Fluorescent beads were bleached using a 561 nm laser at 100% power until the fluorescence level resided within the camera dynamic range. Both fluorescent tags were imaged in EPI mode (laser perpendicular to the sample) with an exposure time of 50 ms and an EMCCD gain of 200 (the experiment length depends of the protein abundance). 15 to 27 flagella were imaged per cell line. Peak fit, lateral drift and channel alignment were performed with ZEN Black (ZEN Black 2.1 SP3, version 14.0.4.201, 64-bit), while the follow up localization and colocalization analysis was performed via in house Fiji version 2.1.0 (Fiji is Just Image J[78]) and R scripts (http://www.R-project.org/) (https://github.com/GiacomoGiacomelli/Carp3-Co-localization-PALM). R scripts were written in RStudio (version 1.1.456) (RStudio_Team 2016) and run on R version 3.6.3 (2020-02-29) (R_Core_Team 2020). For colocalization analysis, we used the coordinate-based colocalization (CBC) method of Malkusch, et al.[43]. The CBC algorithm calculates a colocalization value for each molecule while taking into account the spatial distribution of the two populations within a user determined radius ($R_{max}$). All CBC values were calculated for six different $R_{max}$ (50, 100. 200, 300, 400, 500 nm) and radius intervals of 5 nm (Supplementary Fig. 3a-f). The resulting colocalization parameter varies between +1 (perfectly correlated / high probability of colocalization), through 0 (non-correlated / low probability of colocalization) to −1 (anti-correlated / this value is more difficult to interpret)[79,80]. As CBC values are expression of the relative distribution of the two protein populations, the values are influenced by the detection efficiency of each fluorescent tag. We therefore generated a control cell line with one *CARP3* allele fused to PAmCherry and the other *CARP3* allele fused to mNeonGreen. While both alleles were expressed at comparable levels, PALM imaging showed substantially more PAmCherry events compared to mNeonGreen events (Supplementary Fig. 2d). While mNeonGreen is suitable for PALM, it is not photoactivatable and sample pre-bleaching is a necessary step for the collection of single molecule events[81]. This results in permanent loss of a significant portion of the mNeonGreen population, hence altering the distribution of CBC values (Supplementary Fig. 2d). As CARP3 molecules are expected to colocalize with one another, we can use the obtained distribution of CBC values as a positive control for testing the remaining protein combinations. As a negative control, the CBC values' distribution for

non-colocalizing proteins was obtained by simulating two independent Poisson point patterns (Supplementary Fig. 2f) via the "*rpoispp*" command[82]. The simulated point patterns were defined by a localization density ($\lambda$, expressed in localizations/nm²) and an area ("owin" class object – see[82]). Specifically, $\lambda_{PAmCherry}$ (0.00013 loc/nm²) and $\lambda_{mNeonGreen}$ (0.000026 loc/nm²) were chosen as to possess the same localization density as the protein populations from the positive control, while the area confining the simulation are comprised of a rectangular structure of 1 μm width and a total area equal to the total area from the positive control (broad approximation of a straight flagellum). The percentage of CBC values above or equal to 0.5 (mNeonGreen to PAmCherry and vice versa) was determined and compared for increasing mNeonGreen concentrations (25, 50, 100, 200, 400 and 800 molecules/μm₂) and constant PAmCherry concentration at an $R_{max}$ of 300 nm and an interval width of 10 nm (Supplementary Fig. 3g, h). This analysis revealed highly similar values for CBC ≥ 0.5 at the different mNeonGreen concentrations, excluding the necessity of separate negative controls for each fluorescent protein pair. The percentage of CBC values above or equal to 0.5 of CARP3-PAmCherry / ACP1-mNG, CARP3-PAmCherry / calpain1.3-mNG, CARP3-PAmCherry / FLAM8-mNG and CARP3-PAmCherry / CARP3-mNG was compared to the percentage of CBC values above or equal to 0.5 in the negative control ($p < 0.05$, multiple comparison after Kruskal-Wallis, one tailed).

### GFP-trap pull-down

Immunoprecipitation of CARP3-YFP or ESAG4-GFP was performed using a GFP nanobody covalently coupled to magnetic agarose beads (GFP-trap) according to the manufacturer's instructions (Chromotek, Martinsried, Germany). Briefly, $2 \times 10^8$ *T. brucei* cells (for CARP3-YFP pull-down: AnTat 1.1E wild type or CARP3-YFP, BSFs or PCFs, $n = 2$ replicates each; for ESAG4-GFP pull-down: MiTat 1.2 13-90 or ESAG4-GFP, BSFs, $n = 2$ replicates) were harvested by centrifugation, washed twice with serum-free culture medium and lysed in 1 mL lysis buffer (10 mM Tris/Cl pH 7.5; 150 mM NaCl; 0.5 mM EDTA; 0.4% NP-40 (ESAG4-GFP) or 0.5% NP-40 (CARP3-YFP); Roche cOmplete protease inhibitor) for 30 min on ice. Soluble proteins were separated by centrifugation (10 min, 20,000 × *g*, 4 °C) and incubated with GFP-trap beads (25 μL beads slurry) for 1 h at 4 °C on an overhead rotator. Beads were washed 4x with lysis buffer, followed by three washes with 50 mM Tris/Cl pH 8. Bound proteins were either eluted by boiling for 10 min with Laemmli sample buffer for SDS PAGE or beads with bound proteins were subjected to on-bead digest for mass spectrometry analysis. After on-bead digestion with trypsin, digested peptides were separated on an Ultimate 3000 RSLCnano (ThermoFisher) with a gradient from 4 to 40% acetonitrile in 0.1% formic acid over 40 min at 300 nL/min in a 15-cm analytical (75 μm ID home-packed with ReproSil-Pur C18-AQ 2.4 μm from Dr. Maisch). The effluent from the HPLC was directly electrosprayed into a Q Exactive HF instrument operated in data dependent mode to automatically switch between full scan MS and MS/MS acquisition. Survey full scan MS spectra (from m/z 250–1600) were acquired with resolution R = 60000 at m/z 400 (AGC target of $3 \times 10^6$). The ten most intense peptide ions with charge states between 2 and 5 were sequentially isolated to a target value of $1 \times 10^5$ and fragmented at 27% normalized collision energy. Typical mass spectrometric conditions were: spray voltage, 1.5 kV; no sheath and auxiliary gas flow; heated capillary temperature, 250 °C; ion selection threshold, 33000 counts.

Protein identification and quantification (iBAQ) was performed using MaxQuant 1.6.3.4 (CARP3-YFP) or 1.6.14.0 (ESAG4-GFP)[83] with the following parameters: Database, TriTrypDB-42_TbruceiTREU927_AnnotatedProteins (CARP3-YFP pull-down) or TriTrypDB-48_TbruceiLISTER427_AnnotatedProteins (ESAG4-GFP pull-down); MS tol, 10 ppm; MS/MS tol, 20 ppm; Peptide FDR, 0.1; Protein FDR, 0.01 Min. peptide Length, 5; Variable modifications, Oxidation (M);

Fixed modifications, Carbamidomethyl (C); Peptides for protein quantitation, razor and unique; Min. peptides, 1; Min. ratio count, 2. Statistical analysis was performed using Perseus 1.6.7.0[84] with the following workflow: proteins only identified by site, reverse hits or potential contaminants were filtered out. For ESAG4-GFP pull downs, three replicate LC-MS/MS runs of each sample were grouped as technical replicates and the median thereof was used for further calculations. The iBAQ values of the remaining proteins were log₂ transformed and missing values were replaced from normal distribution. Only proteins identified in both pull down replicates were considered for statistical evaluation (two-sided Student's t-test) with an FDR ≤ 0.05. The raw and processed mass spectrometry proteomics data have been deposited to the ProteomeXchange Consortium[85] (http://proteomecentral.proteomexchange.org) via the PRIDE partner repository[86] with the dataset identifiers PXD025398 (CARP3-YFP) and PXD025412 (ESAG4-GFP), respectively.

### Tsetse fly infections and imaging of isolated parasites

**Fly infection experiments for *flam8* KO and in situ *FLAM8* rescue.** As previously described[87], *Glossina morsitans morsitans* tsetse flies were maintained in the Trypanosome Transmission Group's insectarium of the Institut Pasteur at 27 °C with 70% relative humidity. Flies were kept in Roubaud cages and fed through an artificial membrane feeding system with fresh mechanically defibrinated sheep blood three times a week. Batches of 50 teneral males (unfed adults emerged from their puparium since 12 h to 72 h) were allowed to ingest culture differentiated stumpy parasites in SDM79 medium supplemented with 10% fetal calf serum and 10 mM L-glutathione.

Flies were starved for at least 48 h before dissection. Four weeks after infection, all living flies were dissected: salivary glands were first rapidly isolated in a drop of PBS, the whole tsetse alimentary tract was then arranged lengthways and the foregut and proventriculus were physically separated from the midgut in distinct PBS drops. The infection was scrutinized in the posterior and anterior midgut (PMG and AMG), in the cardia or proventriculus (PV) and foregut, as well as in the salivary glands (SG). The relative parasite densities per organ were evaluated by eye scoring under a 40× objective using the following scale: 0 for no parasites, 1 for 1 to 10 parasites, 2 for 10 to 100, 3 for 100 to 1000 and 4 for >1000 parasites per microscopic field, as described in Schuster et al.[11]. Then, tissues were dilacerated to allow parasites to spread in PBS and parasites were recovered and treated for further experiments no more than 15 min after dissection. For immunofluorescence, parasites were rapidly allowed to settle on poly-lysine coated slides until drying. Cells were fixed for 10 s in methanol at −20 °C and re-hydrated in PBS during 10 min. Slides were then incubated for 45 min at 37 °C with anti-FLAM8[48,88] (1:500) in PBS with 0,1% bovine serum albumin (BSA), followed by PBS washes and incubation with Alexa Fluor® 488 AffiniPure Goat Anti-Rabbit (Jackson ImmunoResearch; 1:400). After a 2-hour blocking step with 1% BSA, slides were incubated with the anti-CARP3 (1:150) primary and Cy™5 AffiniPure Goat Anti-Rabbit IgG (Jackson ImmunoResearch; 1:400) secondary antibodies. Slides were then stained with DAPI for visualization of their kinetoplast and nuclear DNA contents and mounted under coverslip with Prolong antifade reagent (Invitrogen). Slides were finally observed with a DMI4000 epifluorescence microscope (Leica) and images were captured with an Orca 03-G camera (Hamamatsu). Pictures were acquired with Micro-Manager 1.4 and prepared with ImageJ 1.8.0 (NIH).

**Antwerp fly infection experiments for *carp3* KO and in situ *CARP3* rescue.** Female OF1 mice of 10–14 weeks (Charles River France) were used for trypanosome infections and were housed in stable groups of compatible individuals at medium-density caged with max. 5 mice per cage. The animals' environment was kept within specifically defined limits: temperature of 19.5–24.5 °C and relative humidity of 45–65%. A

12 h day/ 12 h night cycle was respected. Bloodstream parasites present in the blood of cyclophosphamide-immunosuppressed (Endoxan) mice at 6–7 days post-infection were mixed with fresh defibrinated horse blood to obtain around $10^6$ parasites/ml in the initial tsetse fly blood meal. Freshly emerged tsetse flies (*Glossina morsitans morsitans* – Institute of Tropical Medicine Antwerp colony; 24–48 h after emergence) were fed their first blood meal with bloodstream forms from either the *T. brucei* AnTat 1.1 WT, the *carp3* KO (two independent clones KO1 and KO2) or an in situ *CARP3* rescue clone (resc1), supplemented with 10 mM reduced L-glutathione, as described in[89]. Flies were further fed every 2–3 days on uninfected defibrinated horse blood. At a defined period after infection, all living flies were dissected after at least 48 hours of starvation, and different tissues were examined for parasite presence/density by phase-contrast microscopy. To determine in a first experiment the overall tsetse infection rates for the different cell lines, midgut and salivary glands were examined five weeks after infection. Then, to estimate in more detail the infection rates and the parasite density in the posterior & anterior midgut and proventriculus, flies were dissected three weeks after the initial infective bloodmeal (without the supplement of L-glutathione). Parasite abundance was scored using the aforementioned method of Schuster, et al.[11].

## Compliance with ethical standards
All applicable international, national and institutional guidelines for the care and use of animals were followed. Specifically, the experimental use of the OF1 mice for trypanosome-tsetse fly infection is covered by the ITM Animal Ethics Committee clearances nr. BM2012-6 and VPU2017-1; these approvals were given by the Animal ethics Committee of the Institute of Tropical Medicine Antwerp.

## Structural modeling of AC and CARP3 proteins
Structural modeling of CARP3 (Supplementary Fig. 1a) and ESAG4 (Supplementary Fig. 9, Supplementary data 6, 7) was initially carried out using the ColabFold notebook AlphaFold2_Advanced (https://colab.research.google.com/github/sokrypton/ColabFold/blob/main/beta/AlphaFold2_advanced.ipynb) utilizing the MMseqs2 database for generating multiple sequence alignments (MSA)[90]. Structural models were later confirmed by executing the originally released AlphaFold2 code along with the full sequence databases[39] using an installation on a local high-performing cluster. Although CARP3 is trypanosome-specific, AlphaFold2 retrieved 34 unique sequence homologs for generating the MSA, which is just above the alignment depth cut-off of 30 sequences below which the model accuracy was found to decrease significantly[39]. Models of CARP3-AC complexes shown in Supplementary Fig. 9 (and Supplementary data 6, 7) were generated using AlphaFold Colab (https://colab.research.google.com/github/deepmind/alphafold/blob/main/notebooks/AlphaFold.ipynb), a slightly simplified version of AlphaFold v. 2.1.0 trained on oligomeric protein structures for improved prediction of protein complexes[51]. AlphaFold Colab was also used to produce a model of homo-dimeric ESAG4 bound to CARP3 as shown in Fig. 6e. All superpositioning and figures of protein structures were created in PyMOL (*The PyMOL Molecular Graphics System, version 2.5* Schrödinger, LLC.). The buried surface area in the predicted interface between CARP3 and different ACs was calculated using the PDBePISA webserver provided by the EMBL-EBI (https://www.ebi.ac.uk/msd-srv/prot_int/cgi-bin/piserver).

## Statistics and reproducibility
The social motility phenotypes were reproduced on n independent plates with $n > 20$ for AnTat 1.1 'Munich' WT and *carp3* KO (Fig. 1a), $n = 16$ for *CARP3* knock-down (Fig. 1b), $n = 8$ for CARP3-Ty1 and CARP3(154–498)-Ty1 (Fig. 2m), $n = 5$ for CARP3(1-160)-Ty1 (Fig. 2m), $n = 5$ for CARP3Δ3 (Fig. 2m), $n = 3$ for AnTat 1.1 E 'Paris' and *FLAM8*

rescue (Fig. 3h), $n = 9$ for *flam8* KO (Fig. 3h), $n = 2$ for CpdA/B treatment (Supplementary Fig. 7c).

The Western blots and in-gel fluorescence images shown in Figs. 1a, b, 2b, 5c, d, 6b, d, and Supplementary Figs. 1e-f, 2a, c, 4a, b, 6a, c, 7a, 8a-c, g were performed once, however, protein expression of all cell lines (except Fig. 2b and Supplementary Fig. 4b) was verified on multiple independent gels or blots.

CARP3 pull-downs in Fig. 5c, d were performed once. Cyclic AMP pull-downs in Supplementary Fig. 7a were performed once with 2-AHA-cAMP beads and once with 8-AHA-cAMP beads. GFP trap pull-downs in CARP3-YFP cells shown in Supplementary Fig. 8b, c were performed three times in BSFs and twice in PCFs with two replicates each shown on the Western blot in Supplementary Fig. 8b and analyzed by mass spectrometry (Fig. 5a, b), and one BSF replicate shown on the Western blot in Supplementary Fig. 8c with antibody detection of ESAG4 and CARP3. CARP3 pull-downs in ACP-mNG trypanosome lines shown in Supplementary Fig. 8g were performed n times with $n = 2$ (ACP1), $n = 5$ (ACP3), $n = 3$ (ACP4), and $n = 9$ (ACP5).

For immunofluorescence analyses (IFA), n cells were investigated for the following cells lines: $n > 200$ for CARP3 localization in PCFs (including different cell cycle stages 1K1N, 2K1N, 2K2N) shown in Fig. 1e and Supplementary Fig. 1h, >14 independent experiments; $n > 20$ cells for *carp3* KO cells shown in Figs. 1f, 2 independent experiments; n > 10 cells for CARP3 redistribution at each time point during BSF to PCF development shown in Fig. 1g, one experiment; $n > 100$ cells for ACP1/6-Ty1 shown in Fig. 2a, 2 independent experiments; $n > 50$ cells for CARP3 localization in BSFs shown in Supplementary Fig. 1i, 3 independent experiments; $n > 10$ cells for CARP3 localization in cell lines expressing CARP3-Ty1, CARP3(154–489)-Ty1, CARP3(1-160)-Ty1 or CARP3Δ3 shown in Fig. 2k, l, 2 independent experiments; $n > 40$ cells for CARP3-mCherry and FLAM8-YFP localizations in each BSF cell line of FLAM8-YFP CARP3-mCh, *flam8* KO CARP3mCh, *carp3* KO FLAM8YFP shown in Fig. 3a, 2 independent experiments; $n > 80$ cells for CARP3 localization in *flam8* KO cells shown in Fig. 3b, 3 independent experiments; $n > 20$ cells for FLAM8 and CARP3 localization in each trypanosome stage isolated from different organs of 6 infected tsetse flies shown in Fig. 3i; $n > 50$ cells for ACP1-mNG or untagged control cells shown in Supplementary Fig. 2b, 3 independent experiments; $n > 30$ cells for each of the trypanosome cell lines with mNeonGreen tagged PDEB1, FLAM8 or calpain 1.3 cells shown in Supplementary Fig. 2b, single experiment; $n > 100$ cells for EP procyclin or calflagin expression at each time point and each of three replicates in RBP6 cell lines shown in Supplementary Fig. 6d, e; $n > 50$ cells for CARP3-YFP localization upon CpdA/B treatment shown in Supplementary Fig. 7b, single experiment; $n > 15$ cells for ACP3-, ACP4-, ACP5-mNeonGreen localization in procyclic *T. brucei* 29-13 shown in Supplementary Fig. 8f, single experiment.

## Reporting summary
Further information on research design is available in the Nature Research Reporting Summary linked to this article.

# Data availability
The proteomics datasets are available in the PRIDE partner repository with the dataset identifiers PXD025398 (CARP3-YFP pull down), PXD025412 (ESAG4-GFP pull down), PXD025357 (CARP3 BioID) and PXD025401 (CARP3 RNAi quantitative proteomics). Genome sequence and annotation information for CARP3 (Tb927.7.5340) and AC isoforms ESAG4 (Tb427.BES40.13), ACP1 (Tb927.11.17040), ACP3 (Tb927.7.7470), ACP4 (Tb927.10.13040), ACP5 (Tb927.11.13740), ACP6 (Tb927.9.15660), GRESAG4.1 (Tb927.6.760) and BSAL_05460 was obtained from TriTrypDB (https://tritrypdb.org). The source data underlying Fig. 1a, b, d, 2b, Fig. 4a-f, Fig. 5c, Fig. 6a, b, d, and Supplementary Fig. 1b-g, Supplementary Fig. 4b, Supplementary Fig. 5b, Supplementary Fig. 6a-c, Supplementary Fig. 7a,

Supplementary Fig. 8g, h are provided as source data file. Source data are provided with this paper.

## Code availability

R scripts for colocalization analysis of PALM data are available at (https://github.com/GiacomoGiacomelli/Carp3-Co-localization-PALM).

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

## Acknowledgements

We thank Samuel Dean, Jessica Street and George Githure for plasmids, Nicolai Kolev, Frederic Bringaud, Keith Matthews and Melanie Bonhivers for antibodies, and Frank Schwede (Biolog, Bremen) for cAMP beads. We thank Benoit Vanhollebeke and David Pérez Morga for facilities for AC assay and HEK cell experiments (ULB, IBBM), Harry de Koning and Daniel Tagoe for exchange of information and Aline Crouzols and Robin Schenk for research assistance. The work was supported by a BioNa young scientists award of LMU to S.B., MC-IEF Fellowship PIEF-GA-2013-626034 to M.K.G., IN.WBI excellence fellowship to L.R.V., CNPq/Universal Grant 725 422022/2016-0 to D.S. and LMU core funding to M.Bo. Work on super-resolution imaging was supported by grants from the Deutsche Forschungsgemeinschaft (DFG, grant numbers 268759902 and 443931024) to M.Br. B.R. was supported by the Institut Pasteur, the Institut National pour la Santé et le Recherche Médicale (INSERM), the French Government Investissement d'Avenir programme - Laboratoire d'Excellence "Integrative Biology of Emerging Infectious Diseases" (ANR-10-LABX-62-IBEID) and the French National Agency for Scientific Research (projects ANR-14-CE14-0019-01 EnTrypa and ANR-18-CE15-0012 TrypaDerm).

## Author contributions

S.B., B.R., D.S., J.V.D.A., M.Br., and M.Bo. designed and supervised research; S.B., E.C.-A., G.G., L.R.V., E.L., A.A., M.K.G., J.V.D.A., A.B. performed research; J-W.D., I.F., A.I. performed, analysed or supervised mass spectrometry; S.B., G.G., E.C.-A., L.R.V., A.A., E.L., B.R., M.Br., D.S., J.V.D.A., M.Bo. analyzed data; S.B. and M.Bo. wrote the paper.

## Funding

## Competing interests

The authors declare no competing interests.
