## [Peer Review File · Nature Communications]

A multi-adenylate cyclase regulator at the flagellar tip controls
African trypanosome transmissionREVIEWER COMMENTS

Reviewer #2 (Remarks to the Author):

NOTEWORTHY RESULTS AND SIGNIFICANCE TO THE FIELD:

This very interesting paper examines the 'directed' movement of procyclic trypanosomes in tsetse flies (a phenomenon called SoMo) and the main claims of the paper relate to the architecture of a flagellar tip microdomain containing adenylate cyclase (AC) and trypanosome-specific proteins and linking the loss of proteins from this domain with the loss of SoMo competence. The meticulous dissection of the localization of CARP3 and FLAM8 and their interactions by microscopy and mutation analysis provides compelling evidence of a dependency relationship between CARP3 and FLAM8 for tip localization in procyclic forms. The data is strong on the myristylation-dependent flagellar tip localisation of CARP3. Equally the localisation of FLAM8 was determined with precision and the dependency of CARP3 on FLAM8 is demonstrated convincingly. This establishes dependency relationships of FLAM8 and CARP3 targeting and the SoMo phenotype and provides solid genetic evidence further strengthening the link between SoMo and cAMP signalling. This detailed mechanistic investigation contributes important new and high-quality data towards the long-elusive goal of identifying downstream targets of cAMP in trypanosomatids.

REVISIONS REQUIRED TO SUPPORT CONCLUSIONS AND KEY CLAIMS

The authors note that "the challenge is to integrate this information into a consistent model" (Line 743). I agree and in my opinion, the proposed model, which is very attractive as a working model, is not as strongly supported by the data as the authors claim in this manuscript, the major flaw being the combination of data from BSF and PCF in support of the model without addressing the fact that the proteins under investigation are known to be differentially localized and the SoMo is particular to procyclic forms.

Major points:

1. The data linking CARP3 abundance with AC abundance and activity is strong for BSF and more limited for PCF, yet the main claims of the paper are about AC regulation at the flagellar tip of PCF. Evidence from diverse sources (including this paper) show that both CARP3 localisation and loss-of-function phenotypes for components of the proposed microdomain are different in PCF and PCF. In bloodstream forms CARP3 localises along the flagellum and posterior cell pole, in PCF it is at the flagellar tip – proximity to ACP1 and ACP6. In the discussion (line 577-80) the BioID data is cited as evidence for their model of the complex at the flagellar tip membrane. The BioID proximity labelling was only done in BSF, and it identified a large number of flagellar proteins consistent with its localisation along the flagellum. While the data is consistent with CARP3 being in close proximity to FLAM8 and ACs it does not provide a proximity map of the PCF flagellar tip domain that is directly relevant for the SoMo phenotype seen in CARP3 depleted cells. The data ARE consistent with the maintenance of a complex, independent of a specific domain within the flagellum. Notably, interaction of CARP3 and the BSF-specific AC ESAG4 was even maintained in HEK cells. The BioID and the activity measurements in the HEK cells ARE NOT evidence directly supporting the presence of an AC complex at the flagellar tip that regulates SoMo.

2. Following on from the above, there is also limited direct evidence for the conclusion that CARP3 modulates flagellar tip cAMP levels by stabilizing AC abundance at this tip microdomain. In BSF knockdown of CARP3 resulted in decreased cAMP production and decrease of ESAG4 levels. ESAG4 is only expressed in BSF – it is unclear how is the ESAG4 data is relevant to the procyclic flagellar tip complex and SoMo. Expression data in BSF then showed decrease of six ACs (Fig. 6c) but only one AC (ACP6, Fig 6d) was assessed in the life cycle stage that is relevant for the SoMo phenotype and model of a flagellar tip nanodomain reported in this paper. There is therefore little direct evidence in this paper that shows CARP3 stabilises ACs that are relevant for SoMo. Direct interaction of CARP3 with ACP4 and ACP5 in PCF was shown in Extended data Fig. 7g but their stability in the absence of CARP3 was not assessed. To strengthen key conclusions of this paper, it would be relevant to know what happens to the procyclic ACs that directly interact with CARP3 and to

the other ACPs shown in the model in Fig. 6f: ACP1, 3, 4 and 5. This would clarify whether loss of CARP3 generally destabilises ACs or whether it affects ACs differentially (as suggested by the authors in the discussion, line 764: "CARP3 depletion might result in local compositional imbalance of AC isoforms possibly due to isoform-specific quantitative differences in CARP3-mediated complex stability.")

3. The fly experiments test whether CARP3 or FLAM8 mutants progress through flies. Taken together the data are interpreted as showing that the only defect seen in these mutants is a failure to migrate, their intrinsic competence to differentiate to different cell forms is not affected. Three separate fly infection experiments were done and it was not straightforward to dissect the data and understand which experiments are directly comparable:

(i) Pasteur - FLAM8, control named "parent"

(ii) Antwerp - CARP3, SG and MG assessed (Fig 1), WT 100% infection

(iii) Antwerp - CARP3, PV and MG assessed but not SG (Fig 4), WT only 60% infection

Tsetse fly infections are known to be variable with regards to SG infection rates. The authors should show clearly which data came from the same experimental time course and not split the fly data across different figures.

They should explain the differences between "parent" (Fig. 4c,d) and "WT" (Fig. 4a,b) and make differences between experimental conditions explicit.

They should also comment on the differences between fly infection rates shown in Fig 1 (day 34-36) and Fig 4. (day 28):

- In Fig 1 SG infection rates of WT were close to 60%, in Fig 4 they were only around 10% for the parent of Flam8.

- What explanation is there for the lower MG infection rates for CARP3 KO and FLAM8 KO, compared to WT, in Fig.4a and 4c?

- For the CARP3 KO only day 21 MG and PV data is shown in Fig 4 but no SG data. Did the authors check whether in these infections resulted in colonization of the SG by the WT cells?

- The finding of zero SG infection with CARP3 mutants is clearly striking. However, given the variation of SG infection rates across these experiments, how confident can one be that this result is significantly outside the variation one sees with WT infections?

Clearer data presentation and clarifications within the text would address these issues.

Minor points:

4. The use of the Alpha fold models is nice additional evidence for a conserved interaction between CARP3 and different *T. brucei* ACPs. An AC from the free-living *Bodo saltans* was used as a negative control. Since CARP3 is said to be found only in the genus *Trypanosoma*, a closer relative with a similar digenetic lifecycle might have proved additionally informative, e.g. *Leishmania infantum*.

5. line 370 – "The dynamic localization of FLAM8 and CARP3 in the life cycle suggests a specific functional role at the flagellar tip in the parasite stages migrating in the fly's alimentary tract." The data show a similar dynamic localization pattern for CARP3 and Flam8 but a "specific functional role" does not follow from this localization data, it is for example possible that tip localization serves sequestration in a non-functional state.

6. The statement (line 664) "Both Δ carp3 and Δ flam8 cell lines do not show any growth or flagellar motility phenotypes in culture." This should be changed to: Δ carp3 and Δ flam8 cell lines do not show any growth phenotypes and their swimming velocity is unchanged compared to controls. Also it is worth noting that the velocity measurements for Flam8 and CARP3 mutants are not directly comparable: they were assayed at different densities, in different methyl cellulose concentrations with a different set-up; this may all contribute to the four-fold difference in the mean cell velocity of the included controls (Flam8 experiment: $\sim 2.5\mu\text{m}/\text{sec}$ Ext Fig 5b vs. CARP3 experiment: $\sim 10\mu\text{m}/\text{sec}$ Ext Fig 1b).

7. The functional link between PDE and CARP3 merits consideration and more careful wording of some statements:

- line 100 and following, also discussion line 436: specify that CARP3 was first identified as a protein

conferring resistance to phosphodiesterase inhibitors concentrations that are lethal to *T. brucei* bloodstream forms.

- The authors say "The SoMo-negative phenotype of $\Delta pdeb1$ (expected increase of [cAMP]) is here interpreted as mechanistically distinct". On what basis? CARP3 was identified in a screen of mutants that confer resistance to a lethal dose of PDEB1 inhibitor – i.e. very compelling evidence linking the two proteins functionally in bloodstream forms that should not be ignored.

Reviewer #3 (Remarks to the Author):

In this manuscript the authors are making important steps in unravelling the role of cAMP signalling in key steps during the trypanosome life cycle. Trypanosomes have an expanded array of adenylate cyclases and cAMP signalling likely plays a crucial role in the life of this parasites yet we still know very little about its regulation and signalling routes in the cell.

They present evidence for the interaction of components of this signalling pathway that are starting to build our understanding of its regulation. They show that CARP3 interacts with a number of different adenylate cyclases and that its localisation is dependent on its N-terminal myristoylation and FLAM8. Disruption of this protein lead to changes in SoMo and the ability to establish a salivary gland infection in the fly. Overall, the evidence in the manuscript supports the conclusions the authors make. However, I have a number of issues with terminology and the over-interpretation of certain experiments that should be addressed.

The abstract talks about a cAMP nanodomain but later in the manuscript, the authors use the term microdomain. I'm not sure they have evidence to discuss a nanodomain but moreover how are they defining this domain - is it based on their microscopy and IPs or is it from a concept of cAMP signalling operating through small defined regions of a cell/flagellum? Can the authors clarify as this is not clear to me.

A major conclusion of this work is that they have defined the architecture of this complex. However, the majority of their evidence relies on AlphaFold models which are associated with a probability score but are ultimately models. The other evidence they provide does confirm an interaction between CARP3 and an adenylate cyclase but not to the degree that they are claiming. The mutational evidence of CARP3 in my opinion does not help pin down the interaction domain. If the first 3 residues are deleted of CARP3 it is unable to localise to the flagellum but that it is due to the reduced membrane interaction as there is no myristoylation and therefore the additional deletion of another 150 amino acids from the N-terminus does not help narrow down the AC interaction domain. The first 160 amino acids are sufficient for the tip localisation but is that due to them interacting with FLAM8 rather than an adenylate cyclase? As without FLAM8 CARP3 localises along the length of the flagellum membrane rather than at the tip. This suggests that CARP3 interaction with an adenylate cyclase is not stable without FLAM8. The interaction from AlphaFold is relatively small including only a short region of helix (13-41) near the NT – what is the size of this interface? The authors need to demonstrate directly through more targeted mutational experiments to conclude this is the interaction domain.

The authors have analysed the relative co-localisation of various components in they have identified using the CBC metric as a probability measure for co-localisation. However, I found the presentation and description of this data confusing. Could the authors provide some form of schematic to explain what the CBC plots for example in fig 2d/e represent? In places the output from this metric are perhaps not, what you would expect. The positive control which is CARP3 tagged with two different tags gives 22.2% events >0.5 but this seems low given the same protein is being tagged and that the suggestion of its function is as it works in a complex with ACs and there are 2 CARP3 proteins per AC

dimer, so a higher level of co-localisation may have been expected? Moreover, the value for CARP3 and FLAM8 is higher at 30.1% yet from the images these only overlap at the interface between the two signals. Does this suggest there is an issue when tagging with different tags? Have the authors tried the co-localisation experiments with different combinations of tags? On many of the CARP3-PAmCh images (Extended Fig 2d, 2c) there are particularly bright spots set back just from the tip and these are not seen on the mNG tagged version – is this a consistent phenomenon?

In the western blot presented in extended data fig. 4a the C3-Ty species has a double band yet CARP3 is normally on a single band – is there some form of modification that is now visible with this tag? Moreover, the western in fig. 6a has an odd set of band patterns. For many lanes, the PFRA/C band appears as a triplet not a doublet. Plus in the first pair of RNAi lanes there is a doublet below the PFRA/C band in green running above the CARP3 band. This lower doublet appears to match the size and relative position to the doublet in the C3-Ty western above – which is also present in both the red/green channel. It is difficult to interpret the exact cause of these different bands but does raise some concerns about the expression of the add backs etc.

In figure 3b, the outline of the FLAM8 deletion describes a cell that does not have the expected morphology. The posterior is much longer than would be normally seen in a procyclic cell – now clearly there are healthy cells in the population as in the panel next to this is a FLAM8 deletion cell with a more normal morphology. The extended data shows that the growth of the FLAM8 deletion cells is similar to the parental but has the morphology been affected as this may have a knock on effect on the ability to colonise the tsetse fly. Has the deletion of FLAM8 been studied before?

The authors have pointed out that there is large expanded repertoire of ACs in *T. brucei*; however, I found the latter sections of the manuscript when these were discussed in detail difficult to follow. Perhaps additional tables/information to outline the size and relatedness of the different AC families would help. When the authors discuss ESAG4 what do they mean e.g. what is its GeneID? Is this the example from the expression site so there are multiple versions of this and how related are they? In the proteomics the authors discuss AC groups – is this what I am looking for in terms of the ACs, what is an AC group? How many AC groups are there? The authors use AlphaFold to model the catalytic domain of the different ACs with CARP3 and show they all have similar predicted structures, now later on they describe this as remarkable (line 585). But given all 7 ACs will be modelled based on their relatedness to each other and to the catalytic domain of this enzyme this similarity is likely not surprising. For these 7 different AC isoforms what is the percentage identity between their catalytic domains? If it is very high one would expect them to model with a high degree of similarity. As a control, the authors model the AC of *Bodo saltans* and show it would not complex with CARP3. Are there any ACs in the *T. brucei* genome which would not be predicted to interact with CARP3?

For the majority of the paper the authors examine the effect of CARP3/FLAM8 manipulation in PCFs but then use BSFs for swell analysis – is it not possible to do this in PCFs? It seems odd to switch life cycle stage when all the previous phenotyping has been about dissecting signalling for fly colonisation. Moreover, there is a distinct change in localisation of this complex between life cycle stages, which adds an additional complication. The authors could investigate the requirement for FLAM8 on CARP3 function in the BSF as one would expect the loss of FLAM8 not to affect the localisation of CARP3 – superficially at least. Are you able to produce FLAM8 deletion line in BSFs does this affect cAMP on swell analysis?

Other points

Have you used the AlphaFold prediction of CARP3 structure to search the structure databases?

On line 331 and in other places you describe CARP3 as de-localised. This seems an odd phrase as CARP3 still have a localisation it is just no longer at the flagellum tip. I would describe it as a change in localisation.

Why is there a supplementary figure 1 when for every other figure we have extended data?

The discussion feels very long in comparison to the rest of the paper and could be shortened.

Reviewer #4 (Remarks to the Author):

The manuscript describes the identification of a signalling complex at the flagellar tip of the parasite *Trypanosoma brucei*. Although individual components were already known, the work described here presents compelling evidence that some constituents form a signalling complex based on unusual transmembrane adenylate cyclases (AC) that are essential for the in vitro phenomenon of social motility (SoMo) and for a successful transmigration through the tsetse fly resulting in infective metacyclic trypanosomes.

In particular, they characterise a novel component of the putative signalling cascade, CARP3. They postulate that CARP3 serves as a scaffold/anchoring proteins, possibly in conjunction with other proteins, such as calpain 1.3, and directly interacts with several adenylate cyclases, perhaps forming an assembly platform for a signalosome. They also show that another known flagellar tip protein, FLAM8, is required for correct targeting of CARP3. Whether this is a specific effect or based on a general attribute of FLAM8 to target proteins to the flagellar tip is not known. In a previous paper, it was discussed whether FLAM8 might be involved in regulating flagellar length. Although one of the present authors (B. Rotureau) co-authored this paper, this possibility is not discussed in the model presented in this manuscript. In fact, the differential distribution of FLAM8 during life cycle stages argues against its role as a regulator of flagellar length.

The authors use a wide range of complementary techniques (PALM superresolution microscopy, proximity assays and pull-downs) to demonstrate the interaction of components.

The manuscript contains structural predictions about CARP3, homo-dimerization of adenylate cyclases and CARP3-AC interaction. All these structural data are based on modelling using AlphaFold. Although some of the models are supported by biochemical (pull-down, proximity assays) and cell biological data (PALM), no real structures are included and therefore any statements about structural interactions at submolecular level are still speculative.

The authors claim (e.g. line 665) that no flagellar motility phenotype was observed. I don't think this conclusion is valid as appropriate techniques to analyse e.g. flagellar beat patterns and beat frequency have, as far as the manuscript goes, not been employed. The defect in SoMo somehow has to have its basis in an altered microswimming behaviour (e.g. tumbling versus straight line motility). I do not suggest that this should be addressed, but statements to this effect need to be made with caution.

The "Discussion" of the manuscript is way too long (almost ten pages) and not well structured.

It should be noted, that a recently published paper in Nature Communications by Shaw et al. (doi: 10.1038/s41467-022-28293-w) comes to similar conclusions and also identifies CARP3 as a key component. Although the methodology in the published paper is less sophisticated to describe the interaction network, the conclusions (effect on SoMo and establishment of tsetse fly infections, central role of CARP3) are essentially very similar. In addition, the paper by Shaw et al. identifies a potential stimulus to initiate the signalling cascade (change in pH). It is surprising (?) that the findings of the Shaw-paper are only glanced over and key findings are not mentioned at all, although the pre-print manuscript was known to the authors and is cited.

Minor points:

Line 90: Reference required

Line 603: As far as I understand the cited paper (Ref. 53) only a differential lipid composition between the cell body and the flagellum is shown, not between the flagellum and the flagellar tip.

Line 618. The term "scaffold" here and elsewhere in the manuscript is etymologically not quite correct. A scaffold is a transient structure. Here, an anchoring structure or framework seems to better describe

the role of CARP3.

Line 635: A possible role of FLAM8 as a reservoir for CARP3 to increase its local concentration is highly speculative. In general, the "Discussion" contains a lot of very speculative statements that are only indirectly supported by the data.

Résumé

Taken together, the data presented in this manuscript present a significant advance in the field but similar key findings have recently been published by Shaw et. al.

The methodology is very convincing and a huge repertoire of techniques has been employed.

Interpretation of the data is very sound. In comparison to the Shaw paper, this manuscript is much more detailed and adds a lot more details. Therefore, the two datasets complement each other. The authors should have been fairer acknowledging the data of the Shaw paper. Referees familiar with the field are aware even of pre-prints!

The Discussion is far too long and contains too many speculative elements.

REVIEWER COMMENTS

Reviewer #2 (Remarks to the Author):

NOTEWORTHY RESULTS AND SIGNIFICANCE TO THE FIELD:

This very interesting paper examines the 'directed' movement of procyclic trypanosomes in tsetse flies (a phenomenon called SoMo) and the main claims of the paper relate to the architecture of a flagellar tip microdomain containing adenylate cyclase (AC) and trypanosome-specific proteins and linking the loss of proteins from this domain with the loss of SoMo competence. The meticulous dissection of the localization of CARP3 and FLAM8 and their interactions by microscopy and mutation analysis provides compelling evidence of a dependency relationship between CARP3 and FLAM8 for tip localization in procyclic forms. The data is strong on the myristylation-dependent flagellar tip localisation of CARP3. Equally the localisation of FLAM8 was determined with precision and the dependency of CARP3 on FLAM8 is demonstrated convincingly. This establishes dependency relationships of FLAM8 and CARP3 targeting and the SoMo phenotype and provides solid genetic evidence further strengthening the link between SoMo and cAMP signalling. This detailed mechanistic investigation contributes important new and high-quality data towards the long-elusive goal of identifying downstream targets of cAMP in trypanosomatids.

REVISIONS REQUIRED TO SUPPORT CONCLUSIONS AND KEY CLAIMS

The authors note that "the challenge is to integrate this information into a consistent model" (Line 743). I agree and in my opinion, the proposed model, which is very attractive as a working model, is not as strongly supported by the data as the authors claim in this manuscript, the major flaw being the combination of data from BSF and PCF in support of the model without addressing the fact that the proteins under investigation are known to be differentially localized and the SoMo is particular to procyclic forms.

Major points:

1. The data linking CARP3 abundance with AC abundance and activity is strong for BSF and more limited for PCF, yet the main claims of the paper are about AC regulation at the flagellar tip of PCF.

We are aware of the problem of resorting to BSF data when the phenotype is in PCF. The reason was technical, due to higher abundance of individual cyclases in BSF that facilitated biochemical approaches. Proteome analysis of CARP3 knock down in BSF revealed downregulation of several AC isoforms (Fig. 6c), whereas the same approach in PCF (carried out in parallel to the BSF experiment) failed to return AC identifications, probably due to much lower expression of ACs in PCF. We accept the reviewer's criticism that this is unsatisfactory. To overcome the sensitivity problem, we have now carried out new experiments in PCF and generated cell lines expressing mNeonGreen tagged ACP1, 4 or 5 for inducible RNAi of CARP3 (Fig. 6d of the revised manuscript). The results show that abundance of all three ACPs is regulated by CARP3. Whereas ACP1 and 4 levels decrease upon CARP3 knock down (as shown for ACP6 before), ACP5 levels increase. Whether ACP5 levels increase as a direct consequence of CARP3 binding or as a compensatory response to downregulation of the other cyclases cannot be decided by this experiment. We also analysed ACP3, but unfortunately, the ACP3-mNG expression in 3 independent clones was too low for reliable quantification and the data were hence not included.

Evidence from diverse sources (including this paper) show that both CARP3 localisation and loss-of-function phenotypes for components of the proposed microdomain are different in PCF and PCF. In bloodstream forms CARP3 localises along the flagellum and posterior cell pole, in PCF it is at the flagellar tip – proximity to ACP1 and ACP6. In the discussion (line 577-80) the BioID data is cited as evidence for their model of the complex at the flagellar tip membrane.

We are sorry for the mis-understanding, we do not claim that BioID data in BSF support the model of the complex at the flagellar tip in PCF. The model is supported by orthogonal evidence from pull-downs and super-resolution microscopy in PCF and AlphaFold2 structure predictions with PCF-specific AC isoforms. We also now cite as additional evidence two recent studies that show CARP3-ACP3/5 interaction (Shaw et al., 2022; published while our manuscript was under review) and putative CARP3-ACP1 interaction (Velez-Ramirez et al., 2021, a proximity proteomics study in PCF that identified CARP3 as a putative ACP1 interactor). We have changed the wording accordingly (lines 441-447).

The BioID proximity labelling was only done in BSF, and it identified a large number of flagellar proteins consistent with its localisation along the flagellum. While the data is consistent with CARP3 being in close proximity to FLAM8 and ACs it does not provide a proximity map of the PCF flagellar tip domain that is directly relevant for the SoMo phenotype seen in CARP3 depleted cells. The data ARE consistent with the maintenance of a complex, independent of a specific domain within the flagellum.

This is exactly what we conclude, the presence of a core complex architecture in both life cycle stages, independently of its subcellular localization, that is possibly determined by the AC isoforms or additional components like FLAM8 (see discussion in lines 518-522). The presence of a similar complex in BSF is supported by orthogonal evidence from reciprocal pull downs in BSF, BioID and AlphaFold2 structure predictions with BSF-specific AC isoforms. As mentioned above, we have removed the CARP3 BioID in BSF as evidence for the model at the flagellar tip in PCF.

Notably, interaction of CARP3 and the BSF-specific AC ESAG4 was even maintained in HEK cells. The BioID and the activity measurements in the HEK cells ARE NOT evidence directly supporting the presence of an AC complex at the flagellar tip that regulates SoMo.

The experiment in HEK cells supports a core complex architecture independent of the parasite cellular environment. We did not claim that experiments in HEK cells support a model for SoMo regulation at the flagellar tip in PCF.

2. Following on from the above, there is also limited direct evidence for the conclusion that CARP3 modulates flagellar tip cAMP levels by stabilizing AC abundance at this tip microdomain.

In BSF knockdown of CARP3 resulted in decreased cAMP production and decrease of ESAG4 levels. ESAG4 is only expressed in BSF – it is unclear how is the ESAG4 data is relevant to the procyclic flagellar tip complex and SoMo. Expression data in BSF then showed decrease of six ACs (Fig. 6c) but only one AC (ACP6, Fig 6d) was assessed in the life cycle stage that is relevant for the SoMo phenotype and model of a flagellar tip nanodomain reported in this paper. There is therefore little direct evidence in this paper that shows CARP3 stabilises ACs that are relevant for SoMo. Direct interaction of CARP3 with ACP4 and ACP5 in PCF was shown in Extended data Fig. 7g but their stability in the

absence of CARP3 was not assessed. To strengthen key conclusions of this paper, it would be relevant to know what happens to the procyclic ACs that directly interact with CARP3 and to the other ACPs shown in the model in Fig. 6f: ACP1, 3, 4 and 5. This would clarify whether loss of CARP3 generally destabilises ACs or whether it affects ACs differentially (as suggested by the authors in the discussion, line 764: "CARP3 depletion might result in local compositional imbalance of AC isoforms possibly due to isoform-specific quantitative differences in CARP3-mediated complex stability.")

We think that the additional experiments and clarifications described above in reply to point 1 also address all of the concerns expressed by the reviewer in point 2. The new experiments (Fig. 6d of the revised manuscript) show that loss of CARP3 destabilises ACP1 and 4 at the flagellar tip of PCFs (this was included before for ACP6) and increases the level of ACP5, demonstrating relevance for the SoMo phenotype and is fully compatible with our "AC compositional imbalance" model. The only conclusion by analogy from an experiment in BSF concerns the effect of CARP3 on cAMP levels. The measurement is possible in BSF due to ESAG4 abundance, but it is impossible to measure changes in [cAMP] in the tiny tip compartment in whole cells. We have attempted to target several genetic cAMP sensors to the tip, but were unsuccessful with measurements in live moving cells.

3. The fly experiments test whether CARP3 or FLAM8 mutants progress through flies. Taken together the data are interpreted as showing that the only defect seen in these mutants is a failure to migrate, their intrinsic competence to differentiate to different cell forms is not affected. Three separate fly infection experiments were done and it was not straightforward to dissect the data and understand which experiments are directly comparable:

- (i) Pasteur - FLAM8, control named "parent"
 - (ii) Antwerp - CARP3, SG and MG assessed (Fig 1), WT 100% infection
 - (iii) Antwerp - CARP3, PV and MG assessed but not SG (Fig 4), WT only 60% infection
- Tsetse fly infections are known to be variable with regards to SG infection rates. The authors should show clearly which data came from the same experimental time course and not split the fly data across different figures.

The data shown in Fig. 1d (Antwerp - CARP3, SG and MG assessed), Fig. 4a,b (Antwerp - CARP3, PV and MG assessed but not SG) and Fig. 4c,d (FLAM8, MG, PV, SG assessed) are derived from three different and completely independent experiments. They were done at different times and in different laboratories using different protocols as detailed in the methods section. All CARP3 tsetse fly experiments were done in Antwerp with wild type AnTat 1.1 'Munich' as control cell line. FLAM8 tsetse fly experiments were done at the Institut Pasteur Paris with cell line AnTat 1.1 E 'Paris' expressing a fluorescent triple marker (Calvo-Alvarez et al., 2018; Ref. 90 of the revised manuscript) as control. In the legend to Fig. 4, we had indicated the different fly colonies (Antwerp / Paris). We now also added the following sentence in the legend to Fig. 1d: „Institute of Tropical Medicine Antwerp tsetse fly colony “.

They should explain the differences between "parent" (Fig. 4c,d) and "WT" (Fig. 4a,b) and make differences between experimental conditions explicit.

In the legend to Fig. 4 this information is given: infected with *T. brucei* AnTat 1.1 wild type (WT) and AnTat 1.1E 'Paris' (parent); WT and parent is used to make clear that all *flam8* mutant lines express a red fluorescent triple marker (Calvo-Alvarez et al., 2018; Ref. 90 of

the revised manuscript) and hence are not strictly WT. Differences between experimental conditions like the parental trypanosome strains and times of fly dissection were included in the legends. In addition, the CARP3 tsetse infections were initiated with BSF trypanosomes, whereas FLAM8 tsetse infections were started with PCFs; this information is given in the detailed methods section. In the legend to Fig. 4 we now emphasize that the different transmission efficiencies are likely due to the different protocols.

They should also comment on the differences between fly infection rates shown in Fig 1 (day 34-36) and Fig 4. (day 28):

- In Fig 1 SG infection rates of WT were close to 60%, in Fig 4 they were only around 10% for the parent of Flam8.

In the legend to Fig. 1d and Fig. 4 we had indicated the different parental strains, time courses and protocols with addition (or not) of 10 mM L-glutathione in the blood meal. All these factors influence the infection rates (L-glutathione: MacLeod et al., 2007 Parasitology, doi: 10.1017/S0031182007002247; different trypanosome strains: Caljon et al., 2016 and Refs. 63-65 of this manuscript). In the methods section, lines 962-968, we wrote: “To determine in a first experiment the overall tsetse infection rates for the different cell lines, midgut and salivary glands were examined five weeks after infection. Then, to estimate in more detail the infection rates and the parasite density in the posterior & anterior midgut and proventriculus, flies were dissected three weeks after the initial infective bloodmeal (without the supplement of L-glutathione).” To make this clearer, we now added to the legend of Fig. 4a, b: “Note that no L-glutathione was added to the bloodmeal to increase the sensitivity of phenotype detection, resulting in lower midgut infection rates compared to Fig. 1d”.

- What explanation is there for the lower MG infection rates for CARP3 KO and FLAM8 KO, compared to WT, in Fig.4a and 4c?

We did not give a speculative explanation in the manuscript. It is (1) possible that the mutant parasites have a slightly reduced fitness in the tsetse midgut, which is not obvious in culture or (2) that SoMo is impacting on midgut establishment as well as on migration to the salivary glands. No doubt, the latter is the dominant phenotype we have focused on. In the discussion we now added a comment (lines 568-570).

- For the CARP3 KO only day 21 MG and PV data is shown in Fig 4 but no SG data. Did the authors check whether in these infections resulted in colonization of the SG by the WT cells?

No, as not a single trypanosome was detected in any SG of tsetse flies infected with *carp3* KO parasites in the Fig.1d experiment, we focused on phenotypes in PM, AM and PV with a different time course and omission of glutathione (see also our reply to the reviewer's comment above).

- The finding of zero SG infection with CARP3 mutants is clearly striking. However, given the variation of SG infection rates across these experiments, how confident can one be that this result is significantly outside the variation one sees with WT infections?

The experiment in Fig. 1d was done under the most favorable conditions possible for obtaining high infection rates (AnTat1.1 strain, infection with BSF stumpy forms from mice,

5-week time course, addition of L-glutathione) resulting in 100% midgut infection in all infected fly series and 60% SG infection rate in the WT and 75% in the rescue. In contrast, 0/100 flies were SG positive when infected with the *carp3* KO. Based on our extensive experience in tsetse fly infection and its variation we can confidently state that this striking zero SG infection observed for the CARP mutants is far outside the 'normal' variation. We have explained in detail above that the experiments in Fig. 4 were under different experimental conditions and hence variation in absolute SG infection rates between independent experiments is expected.

Clearer data presentation and clarifications within the text would address these issues.

As indicated above, we added few more details to the legends of Fig. 1d, Fig. 4 and in the text. As the data presented in Fig. 1d, 4a, b and 4c, d are indeed derived from completely independent experiments with different questions, we think that the order of presentation is justified.

Minor points:

4. The use of the Alpha fold models is nice additional evidence for a conserved interaction between CARP3 and different *T. brucei* ACPs. An AC from the free-living *Bodo saltans* was used as a negative control. Since CARP3 is said to be found only in the genus *Trypanosoma*, a closer relative with a similar digenetic lifecycle might have proved additionally informative, e.g. *Leishmania infantum*.

We intentionally chose a receptor-type AC from the free-living kinetoplastid *Bodo saltans* as negative control, as we considered the possibility of a secondary, more recent loss of *CARP3* in *Leishmania*. *Leishmania* receptor-type ACs may be phylogenetically too close to *T. brucei* to have lost the CARP3 interaction properties.

Following the reviewer's suggestion, we used AlphaFold2 for complex prediction of CARP3 with three different receptor-type ACs from *Leishmania infantum*. CARP3 interaction is indeed predicted for all three ACs with a very similar binding interface compared to *T. brucei* (compare supplement 1 for reviewers with manuscript Supp Fig. 9A). This is consistent with the presence in *Leishmania* of proteins (annotated as stibogluconate resistance proteins) sharing homology with the N terminus of CARP3, including the two helices predicted to interact with ACs. AlphaFold2 indeed predicts AC interaction via these two helices of a stibogluconate resistance protein from *Leishmania tarentolae* (supplement 2 for reviewers). It is possible that the stibogluconate resistance proteins are alternative AC-interacting proteins in *Leishmania* but we feel that this is beyond the scope of our current manuscript. Importantly, the *Bodo saltans* genome does not harbor any protein with similarity to the CARP3 N terminus, thus validating our choice as negative control.

5. line 370 – "The dynamic localization of FLAM8 and CARP3 in the life cycle suggests a specific functional role at the flagellar tip in the parasite stages migrating in the fly's alimentary tract." The data show a similar dynamic localization pattern for CARP3 and Flam8 but a "specific functional role" does not follow from this localization data, it is for example possible that tip localization serves sequestration in a non-functional state.

We rephrased the sentence: "... a distinct role at the flagellar tip in the parasite stages migrating in the fly's alimentary tract compared to localization along the flagellum in other stages."

6. The statement (line 664) "Both Δ carp3 and Δ flam8 cell lines do not show any growth or flagellar motility phenotypes in culture." This should be changed to: Δ carp3 and Δ flam8 cell lines do not show any growth phenotypes and their swimming velocity is unchanged compared to controls. Also it is worth noting that the velocity measurements for Flam8 and CARP3 mutants are not directly comparable: they were assayed at different densities, in different methyl cellulose concentrations with a different set-up; this may all contribute to the four-fold difference in the mean cell velocity of the included controls (Flam8 experiment: $\sim 2.5\mu\text{m}/\text{sec}$ Ext Fig 5b vs. CARP3 experiment: $\sim 10\mu\text{m}/\text{sec}$ Ext Fig 1b).

We are grateful to this clarification; we cannot directly compare velocity measurements for the two mutant lines performed in different laboratories and have adopted the sentence suggested by the reviewer. Anyway, we only concluded from comparison of mutant to matched WT lines. All technical details and differences are given in the Supplementary Methods section (Supplementary Information p. 23). In addition, we now have added to the legend of Supplementary Fig. 5 the sentence "Note that the ~ 4 -fold difference in the mean cell velocity between control cell lines in Supplementary Fig. 5b and Supplementary Fig. 1b is due to differences in cell densities, methyl cellulose concentrations and a different experimental set-up".

7. The functional link between PDE and CARP3 merits consideration and more careful wording of some statements:

- line 100 and following, also discussion line 436: specify that CARP3 was first identified as a protein conferring resistance to phosphodiesterase inhibitors concentrations that are lethal to *T. brucei* bloodstream forms.

We have added the statements in lines 110-111 of the introduction and lines 654-655 of the discussion.

- The authors say "The SoMo-negative phenotype of Δ pdeb1 (expected increase of [cAMP]) is here interpreted as mechanistically distinct". On what basis? CARP3 was identified in a screen of mutants that confer resistance to a lethal dose of PDEB1 inhibitor – i.e. very compelling evidence linking the two proteins functionally in bloodstream forms that should not be ignored.

We agree that PDEB1 and CARP3 are linked by their impact on cAMP concentrations. However, due to its localization along the flagellum, Δ pdeb1 is expected to disrupt cAMP gradients and increases intracellular [cAMP], whereas Δ carp3 modulates cAMP production specifically at the flagellar tip (see also Lohse et al., 2017, <https://doi.org/10.1371/journal.pone.0174856>, and Bock et al., 2019, <https://doi.org/10.1016/j.cell.2020.07.035> for phenotypes of PDE inhibition in mammalian cells). We show here that localized, spatially controlled cAMP signaling is essential for salivary gland colonization by just removing CARP3 from the flagellar tip in Δ carp3, Δ flam8, or CARP3 Δ 3. Δ pdeb1 is expected to cause more widespread effects, i.e. a less specific phenotype. We would like to emphasize that indeed Δ pdeb1 cells are not able to effectively colonize the ectoperitrophic space of the insect's midgut and die very early during

their development in the fly as reported by Shaw et al., 2019, Nat Commun. This phenotype is clearly different from the $\Delta carp3$ and $\Delta flam8$ phenotypes in tsetse flies that is characterized by only slightly reduced midgut infection rates, reduced proventriculus infection rates and complete absence of parasites from the salivary glands. We have tried to make this clearer in the discussion.

Reviewer #3 (Remarks to the Author):

In this manuscript the authors are making important steps in unravelling the role of cAMP signalling in key steps during the trypanosome life cycle. Trypanosomes have an expanded array of adenylate cyclases and cAMP signalling likely plays a crucial role in the life of this parasites yet we still know very little about its regulation and signalling routes in the cell.

They present evidence for the interaction of components of this signalling pathway that are starting to build our understanding of its regulation. They show that CARP3 interacts with a number of different adenylate cyclases and that its localisation is dependent on its N-terminal myristoylation and FLAM8. Disruption of this protein lead to changes in SoMo and the ability to establish a salivary gland infection in the fly. Overall, the evidence in the manuscript supports the conclusions the authors make. However, I have a number of issues with terminology and the over-interpretation of certain experiments that should be addressed.

The abstract talks about a cAMP nanodomain but later in the manuscript, the authors use the term microdomain. I'm not sure they have evidence to discuss a nanodomain but moreover how are they defining this domain - is it based on their microscopy and IPs or is it from a concept of cAMP signalling operating through small defined regions of a cell/flagellum? Can the authors clarify as this is not clear to me.

We thank the reviewer for this comment and agree that we have not properly used the terms nano- and microdomain. The concept of cAMP compartmentation in so-called microdomains has already been suggested by researchers decades ago. However, only within the last years with the design and development of novel, highly sensitive cAMP FRET sensors, the concept was proven and it was even shown that these domains are within nanometer scale (e.g. Bock et al., 2019: <https://doi.org/10.1016/j.cell.2020.07.035>; Dikolayev et al., 2019: <https://doi.org/10.3389/fphys.2019.01406>). Hence, the term microdomain was replaced by nanodomain by some authors. As we do not have data on [cAMP] at a scale below the size of the flagellar tip, we will use the term “microdomain” throughout the manuscript to avoid any overinterpretation.

A major conclusion of this work is that they have defined the architecture of this complex. However, the majority of their evidence relies on AlphaFold models which are associated with a probability score but are ultimately models.

This comment is in disagreement with reviewer #2 who states: ‘The use of the Alpha fold models is nice additional evidence for a conserved interaction between CARP3 and different *T. brucei* ACPs’. The AlphaFold models do not constitute major evidence but as reviewer #2 points out instead provide additional evidence that outlines the overall architecture of AC-CARP3 complexes. The main experimental evidence for AC-CARP3 complexes is based on orthogonal biochemical methods, namely interaction proteomics (unbiased pull-downs of CARP3 as bait followed by mass spectrometry (Fig. 5a, b); pull-downs of CARP3 as bait followed by detection of specific adenylate cyclases via antibodies or fluorescent tags (Fig. 5c, d; Supplementary Fig. 8c, g); unbiased pull-down of ESAG4 as bait followed by mass spectrometry (Supplementary Fig. 8d), CARP3 proximity proteomics (Supplementary Fig. 8e) and PALM microscopy. AlphaFold2 modeling was used as a complementary approach providing structural detail and visualization. Furthermore, it is now widely accepted that AlphaFold2 adds a completely different high confidence level of protein structure modeling compared to previous approaches.

The other evidence they provide does confirm an interaction between CARP3 and an adenylate cyclase but not to the degree that they are claiming. The mutational evidence of CARP3 in my opinion does not help pin down the interaction domain. If the first 3 residues are deleted of CARP3 it is unable to localise to the flagellum but that it is due to the reduced membrane interaction as there is no myristoylation and therefore the additional deletion of another 150 amino acids from the N-terminus does not help narrow down the AC interaction domain. The first 160 amino acids are sufficient for the tip localisation but is that due to them interacting with FLAM8 rather than an adenylate cyclase? As without FLAM8 CARP3 localises along the length of the flagellum membrane rather than at the tip. This suggests that CARP3 interaction with an adenylate cyclase is not stable without FLAM8.

We agree that we made a mistake in inferring from sufficiency of the first 160 amino acids for tip localization to mapping the AC interaction domain. We apologize for not noticing the over-interpretation. Demonstration of sufficiency of CARP3(1-160) for AC (or FLAM8) interaction would require a pull-down experiment that was not possible for the following reasons: CARP3(1-160) is highly unstable (see Supplementary Fig. 4a) and not detectable by western blot. To achieve eventual stabilization of CARP3(1-160) we would have to try several C-terminal fusions and make new lines. We indicated now that the interaction interface is based on a structural prediction (lines 400-402). However, in the reply to the next point below we provide abundant literature evidence that AlphaFold2 provides high confidence interaction predictions with the PAE and pLDDT scores we achieved. As the precise location of the interaction domain in CARP3 is not affecting any other conclusions in our manuscript we prefer not to further delay publication by an additional pull-down validation. With respect to FLAM8, it is clear that this protein is essential for CARP3 tip localization, but we can clearly see two pools of CARP3 at the tip, one associated with FLAM8 and one associated with the membrane and AC (now indicated in the cartoon in Fig. 6f). The most plausible interpretation (also given in the manuscript) is that FLAM8 is required for transport of CARP3 or unloading or concentrating at the tip. Analysis of a physical interaction between CARP3 and FLAM8 is hampered by the fact that FLAM8 is a highly insoluble cytoskeletal protein, making it impossible to do a pull-down.

The interaction from AlphaFold is relatively small including only a short region of helix (13-41) near the NT – what is the size of this interface? The authors need to demonstrate directly through more targeted mutational experiments to conclude this is the interaction domain.

The interaction interface as predicted by AlphaFold is indeed relatively small and comprises mainly the two α -helices of CARP3 found between residues 13-41. We now calculated the buried surface area in the interface between CARP3 and different ACs using the PDBePISA webserver provided by the EMBL-EBI (https://www.ebi.ac.uk/msd-srv/prot_int/cgi-bin/piserver). The results show that different AC proteins burry 600-700Å² of surface area in the interface with CARP3. While this is indeed a relatively small interface, we note that a comparable dimeric interface for the ARL6-BBS1 complex of only 600Å² result in a stable complex with a dissociation constant of 500nM (Mourao et al., NSMB 2014, PMID: 25402481). While we fully realize and acknowledge that we present computationally predicted models and not experimentally determined structures, we would like to point out that the very high pLDDT scores and low predicted aligned error (PAE) for interface regions strongly suggest a correct overall architecture of the complexes (Evans et al., bioRxiv 2021, <https://doi.org/10.1101/2021.10.04.463034>). Although overall average pLDDT scores are not good measures for determining the correctness of protein complex structure predictions, the

PAE and pLDDT scores for interacting regions are excellent predictors for the correctness of protein complex structure prediction by AlphaFold, which are significantly better than models obtained using protein-protein docking methods (Yin et al., bioRxiv 2021, <https://doi.org/10.1101/2021.10.23.465575>). pLDDT >90 and PAE < 5 Å for interacting region for CARP3 and all ACs depicted in Supplementary Fig. 9 of the revised manuscript mean that complex structure prediction is achieved with very high confidence.

However, we fully recognize that our models do not have the required ‘resolution’ to accurately model residues of the interface, and for exactly this reason we only present C-alpha traces in figures and do not discuss interface-specific residues in the manuscript. Indeed, a recent comparison between AlphaFold generated models used for molecular replacement with experimentally derived structures suggest that although large parts of the models superpose perfectly, more than 15% of all sidechains are in the wrong conformation (assigned a wrong rotamer, see Flower and Hurley, *Protein Science*. 2021;30:728–734), making selection for residues for site-directed mutagenesis difficult. A crystal structure of the complex and detailed mutational analyses is beyond the scope of the current work and more importantly, the precise interaction interface does not add to any of the conclusions of the manuscript. We added the surface area of the interface and a word of caution to indicate that the interface is based on a structural model (lines 400 to 405).

The authors have analysed the relative co-localisation of various components in they have identified using the CBC metric as a probability measure for co-localisation. However, I found the presentation and description of this data confusing. Could the authors provide some form of schematic to explain what the CBC plots for example in fig 2d/e represent? In places the output from this metric are perhaps not, what you would expect. The positive control which is CARP3 tagged with two different tags gives 22.2% events >0.5 but this seems low given the same protein is being tagged and that the suggestion of its function is as it works in a complex with ACs and there are 2 CARP3 proteins per AC dimer, so a higher level of co-localisation may have been expected? Moreover, the value for CARP3 and FLAM8 is higher at 30.1% yet from the images these only overlap at the interface between the two signals. Does this suggest there is an issue when tagging with different tags? Have the authors tried the co-localisation experiments with different combinations of tags? On many of the CARP3-PAmCh images (Extended Fig 2d, 2c) there are particularly bright spots set back just from the tip and these are not seen on the mNG tagged version – is this a consistent phenomenon?

Coordinate based co-localization (CBC) values are an established methodological approach to study co-localization in SMLM imaging. This method has been described extensively in Malkush et al. which is cited in the text when CBC analysis is first mentioned (line 197, Ref. 43 of the revised manuscript). Furthermore, we already provide an extensive explanation of the core concepts involving CBC and CBC evaluation in the material and methods (lines 816-823): “The CBC algorithm calculates a colocalization value for each molecule while taking into account the spatial distribution of the two populations within a user determined radius (R_{max}). All CBC values were calculated for six different R_{max} (50, 100, 200, 300, 400, 500 nm) and radius intervals of 5 nm (Supplementary Fig. 3a-f). The resulting colocalization parameter varies between +1 (perfectly correlated / high probability of colocalization), through 0 (non-correlated / low probability of colocalization) to -1 (anti-correlated / this value is more difficult to interpret)”. Said explanation is referred to in the main text (lines 197-198) to help the reader. Finally, we also discussed the effect that changes in parameters, such as molecule number and R_{max} , have on CBC analysis (Supplementary Fig. 3),

highlighting the robustness of the taken approach and controls. As such, we do not think that another schematic is required.

Concerning the “unexpected” percentages in co-localization, we already discuss how the photophysical properties of mNeonGreen influence the result (lines 830-834) “While mNeonGreen is suitable for PALM, it is not photoactivatable and sample pre-bleaching is a necessary step for the collection of single molecule events. This results in permanent loss of a significant portion of the mNeonGreen population, hence altering the distribution of CBC values (Supplementary Fig. 2d).” In brief, bleaching of mNeonGreen in fixed cells results in the heterogeneous loss (dense protein clusters require longer to bleach compared to small oligomers) of a significant portion of the detectable protein population. Nevertheless, the PAmCherry-mNeonGreen combination is one of the few existing dual color PALM approaches that work reliably. Being aware of these limitations, we performed the due controls (shown in Supplementary Fig. 3) and convincingly demonstrate that the tested protein combinations are characterized by high degree of co-localization within the flagellar tip area. Our co-localization data can though not be treated as a strictly quantitative result since the obtained percentages partially depend on which protein has been tagged with mNeonGreen (high-/low clustering and general abundance). We think that major emphasis toward this issue within the paper is confusing and with no impact on the conclusions we draw; we therefore refer the reader to further literature on the topic in the revised version (Stockmar et al., 2018, Ref 80 of the revised manuscript).

Concerning the general lack of bright mNeonGreen foci compared to PAmCherry in the CARP3-PAmCherry-CARP3-mNeonGreen strain, the same argument as described above applies. Briefly, mNeonGreen foci need bleaching prior imaging, meaning that part of the fluorescent population is destroyed, therefore resulting a decreased yield, which in turn translates to dim/less obvious foci.

In the western blot presented in extended data fig. 4a the C3-Ty species has a double band yet CARP3 is normally on a single band – is there some form of modification that is now visible with this tag?

Most likely the C3-Ty sample on this blot suffers some proteolytic degradation, based on our observation that other lysates of the same cell line did not show double bands. To address the concern, we replaced the Western blot in Extended Fig. 4 with a different blot that shows several clones of each of the investigated cell lines (now Supplementary Fig. 4a of the revised manuscript).

Moreover, the western in fig. 6a has an odd set of band patterns. For many lanes, the PFRA/C band appears as a triplet not a doublet. Plus in the first pair of RNAi lanes there is a doublet below the PFRA/C band in green running above the CARP3 band. This lower doublet appears to match the size and relative position to the doublet in the C3-Ty western above – which is also present in both the red/green channel. It is difficult to interpret the exact cause of these different bands but does raise some concerns about the expression of the add backs etc.

We understand the concern, but interpret the extra PFR bands by some proteolytic degradation of the samples that most likely happened during shipment between labs. The AC assays were done in Belgium and the corresponding lysates for Western blot analysis were collected in Belgium but then shipped to Munich for analysis by two-color fluorescent Western blot. We did not quantify the signals on this Western. Even without accurate quantification, this control shows convincingly that CARP3 is depleted upon inducible RNAi

in the AC assays. The lower degradation products seen in the left replicate in Fig. 6a are also PFR-derived as they are present in the green channel only.

In figure 3b, the outline of the FLAM8 deletion describes a cell that does not have the expected morphology. The posterior is much longer than would be normally seen in a procyclic cell – now clearly there are healthy cells in the population as in the panel next to this is a FLAM8 deletion cell with a more normal morphology. The extended data shows that the growth of the FLAM8 deletion cells is similar to the parental but has the morphology been affected as this may have a knock on effect on the ability to colonise the tsetse fly. Has the deletion of FLAM8 been studied before?

Deletion of FLAM8 in BSF is reported by E. Calvo-Alvarez and B. Rotureau (both coauthors on the current manuscript) in a recent preprint (Calvo-Alvarez et al., 2021 bioRxiv; Ref. 47 of the revised manuscript). No changes in morphology, growth, swimming velocity or parasite fitness were observed. Deletion of FLAM8 in PCF has not been described previously. We agree that the cell shown in Fig. 3b does not have average procyclic morphology. We have replaced the image with a different cell with average morphology, and to convince the reviewer that no biased choice was made, we add an overview microscopy image of CARP3 IFA in FLAM8 KO parasites as supplement 3 for reviewers.

The authors have pointed out that there is large expanded repertoire of ACs in *T. brucei*; however, I found the latter sections of the manuscript when these were discussed in detail difficult to follow. Perhaps additional tables/information to outline the size and relatedness of the different AC families would help.

The size and relatedness of the different AC families are best described and visualized in Salmon et al., 2012a, b (Refs 31 & 33 of the revised manuscript) and Durante et al., 2020 (Ref. 28 of the revised manuscript). We had listed in Table S6 of our submitted manuscript the 64 ACs we could identify by proteomics with information on clusters as defined by Durante et al., 2020; we have now added the full set of ACs described in Durante et al., 2020 to our table that is now shown as Supplementary Data 5 of the revised manuscript.

When the authors discuss ESAG4 what do they mean e.g. what is its GeneID? Is this the example from the expression site so there are multiple versions of this and how related are they?

ESAG4 (ID Tb427.BES40.13, now mentioned in the results section in line 366 of the revised manuscript) is the copy in the active expression site of the strain we used for the respective experiments (Lister 427).

In the proteomics the authors discuss AC groups – is this what I am looking for in terms of the ACs, what is an AC group? How many AC groups are there?

“AC protein groups” is a term from MaxQuant-based analysis of mass spectrometry data: protein groups are identified by the same peptides; this is very common for proteins of a multigene family. The term should not be confused with AC protein clusters that refer to phylogenetic relatedness (see Durante et al., Ref. 28 of the revised manuscript, and Supplementary Data 5 of the revised manuscript).

The authors use AlphaFold to model the catalytic domain of the different ACs with CARP3 and show they all have similar predicted structures, now later on they describe this as remarkable (line 585). But given all 7 ACs will be modelled based on their relatedness to each other and to the catalytic domain of this enzyme this similarity is likely not surprising. For these 7 different AC isoforms what is the percentage identity between their catalytic domains? If it is very high one would expect them to model with a high degree of similarity. As a control, the authors model the AC of *Bodo saltans* and show it would not complex with CARP3. Are there any ACs in the *T. brucei* genome which would not be predicted to interact with CARP3?

We agree with the reviewer that given the high sequence identity, the highly similar predicted complex structures are perhaps not that remarkable. We have thus re-written the sentence to: “The very similar structure predictions of seven different AC isoforms in complex with CARP3 (Supplementary Fig. 9a, b; Supplementary Data 6) suggest that CARP3 may be a pan-cyclase-interactor in trypanosomes.” (lines 450-453 of the revised manuscript). Alignment of the catalytic domains of all 76 full-length ACs (derived from Durante et al, just removed pseudogenes and ACs without catalytic domain) revealed sequence identity between 63-100%. The same analysis for the seven ACs (ACP1, 3, 4, 5, 6, ESAG4, GRESAG4) gave a range of 65-88%, indicating that these 7 ACs are not closer related to each other than any other *T. brucei* ACs (see also cladogram in supplement 4 for reviewers). The *Bodo* AC that we used as negative control shows ~40% identity to the 7 *T. brucei* ACs within their catalytic domain). We did not systematically model CARP3 in complex with all 76 ACs, but given the above comparison it seems likely that most *T. brucei* ACs interact with CARP3. In the manuscript title CARP3 is designated as “multi adenylate cyclase regulator” with some caution, although CARP3 could well be a “pan adenylate cyclase regulator”.

For the majority of the paper the authors examine the effect of CARP3/FLAM8 manipulation in PCFs but then use BSFs for swell analysis – is it not possible to do this in PCFs? It seems odd to switch life cycle stage when all the previous phenotyping has been about dissecting signalling for fly colonisation. Moreover, there is a distinct change in localisation of this complex between life cycle stages, which adds an additional complication. The authors could investigate the requirement for FLAM8 on CARP3 function in the BSF as one would expect the loss of FLAM8 not to affect the localisation of CARP3 – superficially at least. Are you able to produce FLAM8 deletion line in BSFs does this affect cAMP on swell analysis?

In our reply to comment 2 of reviewer 2 we explain that “the only conclusion by analogy from an experiment in BSF concerns the effect of CARP3 on cAMP levels. The measurement is possible in BSF due to ESAG4 abundance, but it is impossible to measure changes in [cAMP] in the tiny tip compartment in whole cells. We have attempted to target several genetic cAMP sensors to the tip, but were unsuccessful with measurements in live moving cells”. In addition, the swell dialysis assay is not possible for PCF, as it is based on the BSF-specific osmotic stress-induced stimulation of cyclase activity, not observed in PCF (Rolin 1996, JBC, DOI: [10.1074/jbc.271.18.10844](https://doi.org/10.1074/jbc.271.18.10844)). The swell dialysis assay in CARP3 RNAi in BSF is included to show that in principle cAMP levels are modulated by the CARP3-AC interaction observed in both life cycle stages. FLAM8 KO in BSF has been reported in Calvo-Alvarez et al., 2021, bioRxiv (Ref. 47 of the revised manuscript). CARP3 does not enter the flagellum in the FLAM8 KO (revised version of Calvo-Alvarez et al., 2021, bioRxiv, will be uploaded soon). Therefore, measuring cyclase activity in BSF FLAM8 KO parasites is not relevant for the current manuscript.

Other points

Have you used the Alphafold prediction of CARP3 structure to search the structure databases?

Yes, we have searched the DALI protein structure comparison server (<http://ekhidna2.biocenter.helsinki.fi/dali/>) for similar structures to the AlphaFold2 CARP3 model. The results are shown in the supplement 5 for reviewers. Proteins with similar structure (scored significant) are annotated as G-protein signaling modulators. The sequence identity to CARP3 is, however, very low (8-15%). The hit with the highest similarity is GPSM2 (G-protein-signaling modulator 2), a regulator of the activity of G α proteins that regulate AC activity in many eukaryotes, but are absent from kinetoplastid genomes. As the biological significance of the finding is currently unclear, we prefer not to mention this in the manuscript.

On line 331 and in other places you describe CARP3 as de-localised. This seems an odd phrase as CARP3 still have a localisation it is just no longer at the flagellum tip. I would describe it as a change in localisation.

We agree and have changed “delocalized” to “a change in localization” or ‘mislocalization’ throughout the manuscript.

Why is there a supplementary figure 1 when for every other figure we have extended data?

This is simply due to the initial formatting of the manuscript for a different Nature family journal from which it was directly transferred. We have now renumbered the Supplementary Figures.

The discussion feels very long in comparison to the rest of the paper and could be shortened.

We have shortened the discussion by 436 words (20%).

Reviewer #4 (Remarks to the Author):

The manuscript describes the identification of a signalling complex at the flagellar tip of the parasite *Trypanosoma brucei*. Although individual components were already known, the work described here presents compelling evidence that some constituents form a signalling complex based on unusual transmembrane adenylate cyclases (AC) that are essential for the *in vitro* phenomenon of social motility (SoMo) and for a successful transmigration through the tsetse fly resulting in infective metacyclic trypanosomes.

In particular, they characterise a novel component of the putative signalling cascade, CARP3. They postulate that CARP3 serves as a scaffold/anchoring proteins, possibly in conjunction with other proteins, such as calpain 1.3, and directs interacts with several adenylate cyclases, perhaps forming an assembly platform for a signalosome. They also show that another known flagellar tip protein, FLAM8, is required for correct targeting of CARP3. Whether this is a specific effect or based on a general attribute of FLAM8 to target proteins to the flagellar tip is not known. In a previous paper, it was discussed whether FLAM8 might be involved in regulating flagellar length. Although one of the present authors (B. Rotureau) co-authored this paper, this possibility is not discussed in the model presented in this manuscript. In fact, the differential distribution of FLAM8 during life cycle stages argues against its role as a regulator of flagellar length.

In previous publications from B. Rotureau's lab (Bertiaux E, *Curr Biol*, 2018, doi: 10.1016/j.cub.2018.10.031; and Calvo-Alvarez, *Cell Microbiol*, 2021, Ref. 60 of the revised manuscript), FLAM8 is used as a descriptive, yet very accurate, marker of flagellum maturation, and it is concluded that FLAM8 is not a regulator of flagellum length. This is compatible with the observed stage-specific FLAM8 distribution, important to establish a PCF-specific signaling domain.

The authors use a wide range of complementary techniques (PALM superresolution microscopy, proximity assays and pull-downs) to demonstrate the interaction of components.

The manuscript contains structural predictions about CARP3, homo-dimerization of adenylate cyclases and CARP3-AC interaction. All these structural data are based on modelling using AlphaFold. Although some of the models are supported by biochemical (pull-down, proximity assays) and cell biological data (PALM), no real structures are included and therefore any statements about structural interactions at submolecular level are still speculative.

We agree with the reviewer that details of molecular interaction at a residue-based level cannot be gleaned with high confidence from AlphaFold models and for this reason we have refrained from making statements about interactions at a residue-based level and we only show C-alpha traces of predicted protein structures in figures, none of which are required for the conclusions of the manuscript. However, we would like to point out that although homology modeling of protein structures has a bad reputation among structural biologists as they often do not improve the template, machine learning methods for structure prediction as implemented in AlphaFold2 are vastly superior and result in correct predictions even with very low sequence identity to proteins of known structure. As we outlined in our reply to reviewer #3 above, our structural predictions come with excellent pLDDT and PAE scores for interface regions, which have been shown to be excellent measures for the quality and correctness of the models.

In addition, several lines of evidence beside AlphaFold modeling corroborate the dimeric state of ACs. AC dimerization was previously shown by Salmon et al., 2012b and Saada et al., 2014.

The authors claim (e.g. line 665) that no flagellar motility phenotype was observed. I don't think this conclusion is valid as appropriate techniques to analyse e.g. flagellar beat patterns and beat frequency have, as far as the manuscript goes, not been employed. The defect in SoMo somehow has to have its basis in an altered microswimming behaviour (e.g. tumbling versus straight line motility). I do not suggest that this should be addressed, but statements to this effect need to be made with caution.

We apologize for the overinterpretation "... no flagellar motility phenotype was observed". Of course, we can only exclude differences in swimming velocity with the methods applied. We changed the sentence to " $\Delta carp3$ and $\Delta flam8$ cell lines do not show any growth phenotypes in culture and their swimming velocity is unchanged compared to controls" (lines 554-555).

The "Discussion" of the manuscript is way too long (almost ten pages) and not well structured.

We have shortened the discussion by 436 words (20 %).

It should be noted, that a recently published paper in Nature Communications by Shaw et al. (doi: 10.1038/s41467-022-28293-w) comes to similar conclusions and also identifies CARP3 as a key component. Although the methodology in the published paper is less sophisticated to describe the interaction network, the conclusions (effect on SoMo and establishment of tsetse fly infections, central role of CARP3) are essentially very similar. In addition, the paper by Shaw et al. identifies a potential stimulus to initiate the signalling cascade (change in pH). It is surprising (?) that the findings of the Shaw-paper are only glanced over and key findings are not mentioned at all, although the pre-print manuscript was known to the authors and is cited.

We did in fact mention key findings of Shaw et al. We described the SoMo phenotype of ACP5 sKO cells in lines 660-661 and the pH taxis during SoMo in line 627 of our revised manuscript. We did not mention CARP3 data, as we did not want to enter a controversial discussion of a preprint under review at the very same journal. Now, that the paper is published, we should do that: (1) the CARP3 localization in Shaw et al. is completely different from our data and incompatible with proximity proteomics from the Hill lab (Velez-Ramirez et al. 2021, Ref. 17 of the revised manuscript). As only our work confirms the localization of fusion proteins with an antibody detecting native CARP3, the localization in Shaw et al. is obviously an artefact of N-terminal tagging (as is the mNeonGreen-CARP3 localization from the TrypTag database); this is well explained by the essential role of N-terminal myristoylation. (2) The tsetse infection phenotype of the *carp3* KO parasite strain used in Shaw et al. (a monomorphic *T. brucei brucei* Lister 427 strain defective in development and not fly transmissible) does not match the tsetse infection phenotype of our *carp3* KO trypanosome strain (a pleomorphic *T. brucei brucei* AnTat 1.1 strain with full developmental competence). They observed a severe midgut infection phenotype by *carp3* KO cells that contrasts with high midgut infection rates of our *carp3* KO strain, that showed compromised proventriculus infection and absence from salivary glands. We were surprised

that Shaw et al. analysed fly infection phenotypes in a strain that is not fly transmissible, particularly as we had presented our fly data already 2020 at the BSP TrypLeish meeting in Granada and Shaw et al. added the fly data to their preprint v2 later, probably upon reviewer's request. We have now discussed the CARP3 results of Shaw et al. in the discussion of our revised manuscript (lines 445-447, 531-535, 570-609).

Minor points:

Line 90: Reference required

We added Anton et al., Cell, 2022 (<https://doi.org/10.1016/j.cell.2022.02.011>)

Line 603: As far as I understand the cited paper (Ref. 53) only a differential lipid composition between the cell body and the flagellum is shown, not between the flagellum and the flagellar tip.

The reviewer likely refers to Sharma, A. I. *et al. Scientific reports* 7, 9105 (2017), doi:10.1038/s41598-017-08770-9, our reference 54. We misinterpreted Fig. 3C as tip enrichment of lipid rafts. We agree that there is no convincing evidence in the reference and have removed our speculation and deleted this part of the discussion.

Line 618. The term "scaffold" here and elsewhere in the manuscript is etymologically not quite correct. A scaffold is a transient structure. Here, an anchoring structure or framework seems to better describe the role of CARP3.

Thanks for the suggestion. As non-native speakers, we were not aware of the subtle difference in meaning. The term "scaffold" has been replaced.

Line 635: A possible role of FLAM8 as a reservoir for CARP3 to increase its local concentration is highly speculative. In general, the "Discussion" contains a lot of very speculative statements that are only indirectly supported by the data.

We clearly observed two separate pools of CARP3 at the tip (as also indicated in Fig. 6f of the revised manuscript) and FLAM8 is essential for tip targeting of CARP3, both supporting our speculation. We changed the sentence to "FLAM8 therefore seems to be required for flagellar transport of CARP3 or unloading or concentration at the tip in PCF, possibly to increase local concentration."

Otherwise, we have shortened the discussion (by 436 words, 20%) and have removed several speculative elements.

Résumé

Taken together, the data presented in this manuscript present a significant advance in the field but similar key finding have recently been published by Shaw et al.

The methodology is very convincing and a huge repertoire of techniques has been employed.

Interpretation of the data is very sound. In comparison to the Shaw paper, this manuscript is much more detailed and adds a lot more details. Therefore, the two datasets complement each other. The authors should have been fairer acknowledging the data of the Shaw paper.

Referees familiar with the field are aware even of pre-prints!

The Discussion is far too long and contains too many speculative elements.

We have now discussed in more detail Shaw et al. 2022 (lines 445-447, 531-535, 570-609 of the revised manuscript). We thank the reviewer for encouraging us to highlight the complementarity, but also important differences in conclusions and interpretation. We have also shortened the discussion (by 436 words, 20%) by removing several speculative elements.

Supplement 1 for reviewers: Superimposed AlphaFold2 predicted protein complex structures of *T. brucei* CARP3 (magenta) with *T. brucei* ACP1 (pink) in comparison with three *Leishmania infantum* ACs (colored in green, cyan and red, respectively). GeneIDs for the *Leishmania* ACs are given below the figure.

Supplement 2 for reviewers: Superimposed AlphaFold2 predicted protein complex structures of *T. brucei* CARP3 (magenta) or *Leishmania tarentolae* stibogluconate resistance protein LtaPh_9903201 (dark blue) with *L. infantum* AC LINF_170007300 (green).

Supplement 3 for reviewers: Indirect immunofluorescence analysis of CARP3 (green) in procyclic *T. brucei* AnTat 1.1E *flam8* KO cells. DNA was stained with DAPI (blue), the axoneme is labeled in red (detected by the antibody mAB25). Left: merge of fluorescent channels; right: merge with phase contrast. Scale bars 10 µm.

Supplement 4 for reviewers: Circular cladogram of *T. brucei* AC catalytic domains. AC isoforms are either labeled with their TriTrypDB entries or with their gene names for ACP1, 3, 4, 5, 6, 427 ESAG4 (Tb427.BES40.13) and GRESAG4.1 (Tb927.6.760). The AC isoforms used for AlphaFold2 protein complex structure modeling with CARP3 are highlighted in bold red font.

Select neighbours (check boxes) for viewing as multiple structural alignment or 3D superimposition. The list of neighbours is sorted by Z-score. Similarities with a Z-score lower than 2 are spurious. Each neighbour has links to pairwise structural alignment with the query structure, and to the PDB format coordinate file where the neighbour is superimposed onto the query structure.

Structural Alignment
 Expand gaps
 3D Superimposition (PV)
 SANS
 PANZ

Summary

No:	Chain	Z	rmsd	lali	nres	%id	PDB	Description
[ ]	1:	5a6c-B	11.6	3.9	192	369	11	PDB MOLECULE: G-PROTEIN-SIGNALING MODULATOR 2, AFADIN;
[ ]	2:	3sf4-B	11.6	4.2	196	350	11	PDB MOLECULE: G-PROTEIN-SIGNALING MODULATOR 2;
[ ]	3:	3q15-B	11.5	3.3	185	332	13	PDB MOLECULE: RESPONSE REGULATOR ASPARTATE PHOSPHATASE H;
[ ]	4:	4wnf-A	11.5	4.2	191	255	11	PDB MOLECULE: G-PROTEIN-SIGNALING MODULATOR 2;
[ ]	5:	5a7d-G	11.5	3.8	192	346	13	PDB MOLECULE: PINS;
[ ]	6:	5a7d-E	11.5	3.9	193	352	13	PDB MOLECULE: PINS;
[ ]	7:	3sf4-A	11.3	4.4	195	355	11	PDB MOLECULE: G-PROTEIN-SIGNALING MODULATOR 2;
[ ]	8:	4wnd-A	11.3	4.1	194	329	12	PDB MOLECULE: G-PROTEIN-SIGNALING MODULATOR 2;
[ ]	9:	4ila-A	11.2	3.7	181	274	10	PDB MOLECULE: RESPONSE REGULATOR ASPARTATE PHOSPHATASE I;
[ ]	10:	6hc2-K	11.2	4.3	194	366	11	PDB MOLECULE: G-PROTEIN-SIGNALING MODULATOR 2;
[ ]	11:	4g2v-A	11.1	4.4	193	321	11	PDB MOLECULE: G-PROTEIN-SIGNALING MODULATOR 2;
[ ]	12:	4als-B	11.1	4.0	194	342	13	PDB MOLECULE: PARTNER OF INSCUTEABLE;
[ ]	13:	3sf4-C	11.1	4.2	191	346	12	PDB MOLECULE: G-PROTEIN-SIGNALING MODULATOR 2;
[ ]	14:	4apo-B	11.0	2.8	148	154	11	PDB MOLECULE: AH RECEPTOR-INTERACTING PROTEIN;
[ ]	15:	6px0-A	11.0	1.9	133	135	15	PDB MOLECULE: ARYL-HYDROCARBON-INTERACTING PROTEIN-LIKE 1;
[ ]	16:	4wne-A	11.0	4.4	192	326	11	PDB MOLECULE: G-PROTEIN-SIGNALING MODULATOR 2;
[ ]	17:	6hc2-M	11.0	3.7	177	366	10	PDB MOLECULE: G-PROTEIN-SIGNALING MODULATOR 2;
[ ]	18:	6hc2-I	10.9	3.7	175	367	15	PDB MOLECULE: G-PROTEIN-SIGNALING MODULATOR 2;
[ ]	19:	4aif-A	10.9	2.6	142	144	11	PDB MOLECULE: AH RECEPTOR-INTERACTING PROTEIN;
[ ]	20:	4aif-B	10.8	2.5	142	144	11	PDB MOLECULE: AH RECEPTOR-INTERACTING PROTEIN;
[ ]	21:	3ro2-A	10.8	4.4	193	328	11	PDB MOLECULE: G-PROTEIN-SIGNALING MODULATOR 2;
[ ]	22:	6hc2-S	10.8	3.7	175	362	15	PDB MOLECULE: G-PROTEIN-SIGNALING MODULATOR 2;
[ ]	23:	6hc2-U	10.8	3.9	180	366	10	PDB MOLECULE: G-PROTEIN-SIGNALING MODULATOR 2;
[ ]	24:	6hc2-E	10.8	3.8	180	366	10	PDB MOLECULE: G-PROTEIN-SIGNALING MODULATOR 2;
[ ]	25:	4wng-A	10.7	4.4	192	330	11	PDB MOLECULE: G-PROTEIN-SIGNALING MODULATOR 2;
[ ]	26:	3q15-A	10.7	4.5	190	365	11	PDB MOLECULE: RESPONSE REGULATOR ASPARTATE PHOSPHATASE H;
[ ]	27:	6hc2-C	10.6	3.9	188	367	11	PDB MOLECULE: G-PROTEIN-SIGNALING MODULATOR 2;
[ ]	28:	5a6c-A	10.6	4.0	185	363	12	PDB MOLECULE: G-PROTEIN-SIGNALING MODULATOR 2, AFADIN;
[ ]	29:	6hc2-O	10.6	3.9	188	365	11	PDB MOLECULE: G-PROTEIN-SIGNALING MODULATOR 2;
[ ]	30:	4als-A	10.6	3.9	174	346	9	PDB MOLECULE: PARTNER OF INSCUTEABLE;
[ ]	31:	4apo-A	10.5	2.6	140	142	11	PDB MOLECULE: AH RECEPTOR-INTERACTING PROTEIN;
[ ]	32:	5a7d-F	10.5	3.8	194	346	13	PDB MOLECULE: PINS;
[ ]	33:	5a7d-C	10.4	3.8	193	348	13	PDB MOLECULE: PINS;
[ ]	34:	6hc2-Q	10.4	4.3	187	367	9	PDB MOLECULE: G-PROTEIN-SIGNALING MODULATOR 2;
[ ]	35:	4jhr-B	10.4	3.9	180	289	8	PDB MOLECULE: G-PROTEIN-SIGNALING MODULATOR 2;
[ ]	36:	4jhr-A	10.4	3.9	180	290	8	PDB MOLECULE: G-PROTEIN-SIGNALING MODULATOR 2;
[ ]	37:	5a7d-I	10.3	4.1	187	346	11	PDB MOLECULE: PINS;
[ ]	38:	5o01-A	10.3	4.0	184	254	9	PDB MOLECULE: BKLC (BACTERIAL KINESIN-LIGHT CHAIN-LIKE);

Supplement 5 for reviewers: Proteins with similar structures to the AlphaFold2 predicted structure for CARP3 retrieved from the DALI protein structure comparison server.

REVIEWERS' COMMENTS

Reviewer #2 (Remarks to the Author):

The authors addressed my major comments in detail and clarified my most important questions. Specifically, the additional experiment included in the revised manuscript showing that ACP1, 4 and 6 levels are reduced in PCF upon CARP3 knockdown strengthens the paper considerably. While I appreciate that technical constraints dictated the choice of life cycle stage for in vitro studies, the paper's key claim and model (Fig 6f) are about the role of the CARP3-AC complex in the insect vector stage. The newly added data in Fig 6d provides important evidence to support this model, which was missing from the original submission.

The discussion of the paper by Shaw et al. is also a valuable addition, especially since some discrepancies exist between findings. Just one point of clarification: in lines 524-5 it says "They observed a severe midgut infection phenotype that contrasts with high midgut infection rates of our *carp3* KO strain". I find this wording ambiguous as it is unclear if Shaw et al. saw higher or lower parasite burdens. In fact, Shaw et al. observed significantly lower midgut infection rates for their CARP3 KO.

Overall this comprehensive study provides strong converging evidence from multiple approaches for a CARP3-AC complex at the tip of the PCF flagellum and its dependence on Flam8, and it demonstrates the importance of this complex for the migration behaviour of trypanosomes.

Reviewer #3 (Remarks to the Author):

In this revision the authors have provided substantial further evidence for the importance of CARP3 for the regulation of adenylate cyclase abundance in PCF, confirming its role as part of a complex of proteins that regulate the ability of the parasite to complete its life cycle in the fly.

Overall, the authors have responded to my comments well and have addressed my concerns. In addition, their responses to the other reviewers appear comprehensive.

I am uncomfortable with their use of the word architecture in the manuscript, which in my opinion implies that they are describing the position and interactions of each of the components in the complex. However, most of the evidence for architecture is from AlphaFold which despite the high level of confidence is still a model and as the authors themselves say is not accurate enough to base mutational studies on. They do have support from other approaches but this mainly just confirms that these proteins interact as a complex, not which bits of the proteins are important for that. The only additional architecture evidence is the truncation study showing that the ACs interact through the cyclase domain. I suspect I may be on the wrong side of history but I think the bar for describing architecture in an important paper such as this is higher than that provided.

A minor point is that there is a mix of KO and delta used to describe the null mutants throughout the manuscript.

Reviewer #4 (Remarks to the Author):

In general, the authors have satisfactorily addressed the criticism raised in my initial review of the manuscript, although I still think the Discussion is too long, but I leave this at the editors' discretion. When reading the manuscript again, I pondered about a few experimental details and interpretation of some of the data. The authors may or may not consider this in their manuscript.

In the tsetse fly several motility-dependent events occur. There is, on one hand, the migration of cells

through the different anatomical regions of the fly and, on the other hand, extensive swarm- like swimming e.g. in the ectoperitrophic lumen in the midgut. It is likely that both phenomena are linked and essential for successful life cycle progression.

There have been some interesting papers about collective motility of *Chlamydomonas* algae and the interpretation of the observed swarm behaviour is, that factors of the external environment, such as dimensions and curvature of the confinement, are the major determinants of collective swimming behaviour (Cammann, J. et al. Emergent probability fluxes in confined microbial navigation. *Proceedings of the National Academy of Sciences* 118, (2021); Ostapenko, T. et al. Curvature-Guided Motility of Microalgae in Geometric Confinement. *Phys Rev Lett* 120, 068002 (2018). It might be worth including these contrasting (but not mutually exclusive) views (extrinsic vs. intrinsic cues) somewhere in the Discussion.

A few minor points that might need clarifying:

Do ACP proteins also relocalise in BSF cells? Also, is the BSF-localisation of FLAM8 known? If it is essential for CARP3 localisation it would be surprising and informative if this function does not apply in BSF cells.

In order to generate the YFP-tagged FLAM8 cells procytic cells rather than the differentiation-competent, pleomorphic cells have been used. Why was this done this way, rather than using pleomorphic cells throughout? In lines 283-291 they describe the localisation of FLAM8 for the tsetse fly stages, but don't mention anything about BSF cells. I am not suggesting to do these experiments, but to state that it is either not known or mention what is known about FLAM8 in BSF.

Line 449: As far as I know dual myristoylation/palmitoylation requires a cysteine at position three (MGC...), which CARP3 doesn't have and therefore speculation as to this PTM are probably unnecessary.

REVIEWERS' COMMENTS

Reviewer #2 (Remarks to the Author):

The authors addressed my major comments in detail and clarified my most important questions. Specifically, the additional experiment included in the revised manuscript showing that ACP1, 4 and 6 levels are reduced in PCF upon CARP3 knockdown strengthens the paper considerably. While I appreciate that technical constraints dictated the choice of life cycle stage for in vitro studies, the paper's key claim and model (Fig 6f) are about the role of the CARP3-AC complex in the insect vector stage. The newly added data in Fig 6d provides important evidence to support this model, which was missing from the original submission.

The discussion of the paper by Shaw et al. is also a valuable addition, especially since some discrepancies exist between findings. Just one point of clarification: in lines 524-5 it says "They observed a severe midgut infection phenotype that contrasts with high midgut infection rates of our *carp3* KO strain". I find this wording ambiguous as it is unclear if Shaw et al. saw higher or lower parasite burdens. In fact, Shaw et al. observed significantly lower midgut infection rates for their CARP3 KO.

We agree with the reviewer that the wording was ambiguous and changed the sentence in lines 527-530 (of the revised manuscript) to: "They observed strongly reduced midgut infection rates that contrast with high midgut infection rates of our *carp3* KO strain"

Overall this comprehensive study provides strong converging evidence from multiple approaches for a CARP3-AC complex at the tip of the PCF flagellum and its dependence on Flam8, and it demonstrates the importance of this complex for the migration behaviour of trypanosomes.

Reviewer #3 (Remarks to the Author):

In this revision the authors have provided substantial further evidence for the importance of CARP3 for the regulation of adenylate cyclase abundance in PCF, confirming its role as part of a complex of proteins that regulate the ability of the parasite to complete its life cycle in the fly.

Overall, the authors have responded to my comments well and have addressed my concerns. In addition, their responses to the other reviewers appear comprehensive.

I am uncomfortable with their use of the word architecture in the manuscript, which in my opinion implies that they are describing the position and interactions of each of the components in the complex. However, most of the evidence for architecture is from AlphaFold which despite the high level of confidence is still a model and as the authors themselves say is not accurate enough to base mutational studies on. They do have support from other approaches but this mainly just confirms that these proteins interact as a complex, not which bits of the proteins are important for that. The only additional architecture evidence is the truncation study showing that the ACs interact through the cyclase domain. I suspect I

may be on the wrong side of history but I think the bar for describing architecture in an important paper such as this is higher than that provided.

We changed 'architecture' to 'composition' in abstract and introduction. We prefer to keep the word 'architecture' in the discussion (lines 425-427 of the revised manuscript), but changed the sentence to 'In this work, we propose a novel architecture of a cAMP signaling complex essential for successful arthropod host-parasite interaction and hence transmission of trypanosomes.'

A minor point is that there is a mix of KO and delta used to describe the null mutants throughout the manuscript.

We changed it to knock-out or KO throughout the manuscript.

Reviewer #4 (Remarks to the Author):

In general, the authors have satisfactorily addressed the criticism raised in my initial review of the manuscript, although I still think the Discussion is too long, but I leave this at the editors' discretion. When reading the manuscript again, I pondered about a few experimental details and interpretation of some of the data. The authors may or may not consider this in their manuscript.

In the tsetse fly several motility-dependent events occur. There is, on one hand, the migration of cells through the different anatomical regions of the fly and, on the other hand, extensive swarm-like swimming e.g. in the ectoperitrophic lumen in the midgut. It is likely that both phenomena are linked and essential for successful life cycle progression.

There have been some interesting papers about collective motility of *Chlamydomonas* algae and the interpretation of the observed swarm behaviour is, that factors of the external environment, such as dimensions and curvature of the confinement, are the major determinants of collective swimming behaviour (Cammann, J. et al. Emergent probability fluxes in confined microbial navigation. *Proceedings of the National Academy of Sciences* 118, (2021); Ostapenko, T. et al. Curvature-Guided Motility of Microalgae in Geometric Confinement. *Phys Rev Lett* 120, 068002 (2018). It might be worth including these contrasting (but not mutually exclusive) views (extrinsic vs. intrinsic cues) somewhere in the Discussion.

A few minor points that might need clarifying:
Do ACP proteins also relocalise in BSF cells?

ACP proteins were identified as procyclic-enriched or procyclic-specific adenylate cyclases (Saada et al., 2014, Ref. 18 of the revised manuscript) and hence, their localization in BSFs has not been investigated.

Also, is the BSF-localisation of FLAM8 known? If it is essential for CARP3 localisation it would be surprising and informative if this function does not apply in BSF cells.

The localization of FLAM8 during trypanosome stage differentiation has recently been described by Calvo-Alvarez et al., 2021 (*Cell Microbiol*; Ref. 48 of the revised manuscript).

In BSFs, “FLAM8 was not only present at the distal tip of the flagellum, but it was also distributed along the entire flagellum length, with an enrichment in the proximal half of the flagellum” (Calvo-Alvarez et al., 2021, Cell Microbiol; Ref. 48 of the revised manuscript). Deletion of FLAM8 in BSF is reported by E. Calvo-Alvarez and B. Rotureau (both coauthors on the current manuscript) in a recent preprint (Calvo-Alvarez et al., 2021 bioRxiv; Ref. 47 of the revised manuscript). CARP3 does not enter the flagellum in the FLAM8 KO (revised version of Calvo-Alvarez et al., 2021, bioRxiv, will be uploaded soon).

In order to generate the YFP-tagged FLAM8 cells procyclic cells rather than the differentiation-competent, pleomorphic cells have been used. Why was this done this way, rather than using pleomorphic cells throughout?

This was probably a misunderstanding. All cell lines with tagged CARP3, FLAM8, calpain 1.3 or PDEB1 were generated in pleomorphic BSF cells, i.e. also FLAM8-YFP cells were generated in BSFs, followed by differentiation to PCFs in culture. In order to make this clearer, we added this information to the Supplementary Methods section of ‘C-terminal in situ tagging of FLAM8 with YFP or mNeonGreen: For C-terminal in situ tagging of FLAM8 with YFP, *T. brucei* AnTat 1.1 BSFs were transfected with p3329.FLAM8¹⁴ followed by selection with 0.1 µg/mL puromycin.’

In lines 283-291 they describe the localisation of FLAM8 for the tsetse fly stages, but don't mention anything about BSF cells. I am not suggesting to do these experiments, but to state that it is either not known or mention what is known about FLAM8 in BSF.

We added the previously described localization of FLAM8 in BSFs to the results section (lines 291-293): The CARP3 and FLAM8 localization pattern in salivary gland stages was similar to that in bloodstream forms (Fig. 1g, Supplementary Fig. 1i; FLAM8 in BSFs see ⁴⁸)

Line 449: As far as I know dual myristoylation/palmitoylation requires a cysteine at position three (MGC...), which CARP3 doesn't have and therefore speculation as to this PTM are probably unnecessary.

It is true that CARP3 does not have a palmitoylatable cysteine at position three. We added this to the sentence as follows: “CARP3 has no palmitoylatable cysteine at position three and has not been identified in a palmitoyl proteome“